

# Global Disaster Risk Assessment from Emergency Events Database (2013-2023)

Qingzhao Kong[1], Erqi Zhu[1]

[1]Department of Disaster Mitigation for Structures, Tongji University, Shanghai 200092, China

*Correspondence to*: Qingzhao Kong (qkong@tongji.edu.cn)

**Abstract.** This paper defines an Accumulated Risk Index (ARI) to quantify regional risk levels in a global perspective utilizing Emergency Events Database (EM-DAT) from 2013 to 2023. Building on the World Risk Index (WRI) released from World Risk Report 2023, the ARI focuses more on major events caused by multiple natural hazards within 5°×5° latitude-longitude grids, and calculates their accumulative effects during the period. ARI results are presented in two forms including normalized global maps and index rankings of each grid to identify high-risk areas that warrant increased

attention. To provide a data foundation for subsequent research at smaller scales, this paper integrates and supplements existing remote sensing images within high-risk areas, publishing them as an open-source 3H Dataset partitioned by grids after standardizing their formats. The unified imagery establishes a valuable resource for deeper insights and more precise analyses in future disaster risk research of developing countries.

## 1 Introduction

In association with the compounding effects of climate change and human activities, natural hazards demonstrate a more frequent and intense trend in recent years (Yarveysi et al., 2023; Chamberlain et al., 2024; Wang et al., 2017; Hussain et al., 2023; Taghizadeh-Hesary et al., 2021; Wen et al., 2023; Chitondo et al., 2024; Coronese et al., 2019). Moreover, natural hazards are not isolated, but exhibit clustering features in both time and space dimensions, which indirectly reflect accumulative socio-economic impacts (Ridder et al., 2020; Hufschmidt et al., 2005). It is regrettable that the impacts are

also closely correlated with both the frequency and intensity of disaster events. Particularly in developing countries, major events triggered by multiple natural hazards interact with socio-economic vulnerability, resulting in severe human casualties, substantial economic losses, and heightened social unrest (Koks et al., 2019; Rentschler et al., 2022; Baranowski et al., 2020). Consequently, the escalating risk of major disaster events poses a severe challenge to global sustainable development, which underscores the necessity for comprehensive risk assessments.

There is a rapidly increasing recognition on the need for disaster risk assessments, accompanied by growing dialogue (Koks et al., 2019; Ward et al., 2020). In recent years, a large volume of research has been dedicated to assessing the risks of individual hazard or single region (Ridder et al., 2020; Koks et al., 2019; Julià and Ferreira, 2021). However, owing to the inherent complexity and methodological challenges involved, assessing the probable impacts of multiple natural hazards on a global scale has yet to be widely adopted in mainstream practices (Ward et al., 2020; Julià and Ferreira, 2021;

Gallina et al., 2016), with only initiatives promoted by renowned organizations such as the United Nations Development Programme (UNDP) being exceptions. As summarized in Table 1, representative global multi-disaster risk assessments are compiled, with the following aspects of emphasis that warrant further consideration and analysis:

1) Data timeliness: The frequency and intensity of natural hazards have shown a significant upward trend in recent years due to factors such as climate change (Yarveysi et al., 2023; Hussain et al., 2023; Taghizadeh-Hesary et al., 2021; Wen et

al., 2023; Chitondo et al., 2024; Coronese et al., 2019). Among major indices, only WRI, CRI and INFORM cover disaster statistics in the past 10 years. Furthermore, CRI has stopped updating after 2021 due to data supply issues.

2) Assessment unit: Most representative indices use countries as their minimum assessment unit, with only Hotspots, TRI, and MhRI being exceptions. While the country-level approach facilitates access to reliable socio-economic data as assessment indicators, it is too coarse compared to the actual geographic scope of natural hazards (Shi and Kasperson,



2015; Dilley, 2005; Paprotny et al., 2018). This national-level zonation cannot effectively reflect the spatial distribution of disasters (Mucke et al., 2011) and is poorly suited for disaster research based on remote sensing technology.

3) Disaster type: Several representative indices cover a limited range of natural hazards. For instance, DRI only considers four types with high mortality rates (UNDP, 2004), while CRI focuses exclusively on climate disasters (Eckstein et al., 2021). Some indices, such as INFORM, also incorporate human-made hazards like violent conflicts (Marin-Ferrer et al.,

2017).

4) Assessment indicator: While risk assessments include objective indicators such as population distribution and hazard frequency/intensity, most indicators are subjectively selected (Yarveysi et al., 2023; Dilley, 2005; UNDP, 2004; Marin et al., 2021). Therefore, the reliability of assessment results primarily depends on the comprehensiveness of subjective considerations and chosen indicators. The approaches vary significantly among different indices: from DRI, which only

considers death risk as a single dimension (UNDP, 2004), to WRI, which comprehensively evaluates social and economic aspects using 100 different indicators (Frege et al., 2023; Mucke et al., 2022).

Furthermore, in the implementation of disaster risk assessments, remote sensing technology is widely regarded as a critical source of information and data. Particularly since the advent of research-oriented satellite systems and sensors, it has been extensively used for acquiring spatial information and analyzing distribution data (Pittore et al., 2017; Tronin, 2009; Geiß

and Taubenböck, 2012). Disaster risk assessment relies heavily on the statistical pertaining to exposure and vulnerability. However, data availability and acquisition costs significantly limit the broader application of remote sensing technology in disaster risk research (Geiß and Taubenböck, 2012; Elliott et al., 2016; Rathje and Adams, 2008; Manfré et al., 2012), especially for the developing countries (Herold and Sawada, 2012). Furthermore, there is an inherent trade-off between spatial resolution and coverage area in remote sensing data: higher spatial resolution typically entails a smaller coverage

area (Pittore et al., 2017; Geiß and Taubenböck, 2012; Rathje and Adams, 2008). This constraint restricts the practical application of remote sensing technology in regions with limited financial resources or limited access to high-resolution imagery, making thorough disaster risk assessments more difficult.

Based on these observations, this paper proposes a global multi-disaster risk assessment index that focuses on the accumulative effects of major disaster events worldwide from 2013 to 2023. The index uses 5°×5° latitude-longitude grids

as its basic unit and incorporates WRI as a weight coefficient. By combining these two different assessment units, this index comprehensively integrates both historical patterns and recent conditions of natural hazards. Moreover, this paper integrates and supplements existing sub-meter resolution visible spectral remote sensing images within high-risk areas based on the index, standardizes their formats and publishes them as an open-source dataset for subsequent research at smaller scales.





**Table 1: Representative results of global multi-disaster risk assessment (Shi and Kasperson, 2015; Dilley, 2005; UNDP, 2004; Eckstein et al., 2021; Marin-Ferrer et al., 2017; Frege et al., 2023).**

| Name | Source | Publication time | Statistic year | Assessment unit | Natural hazard type | Unified parameter | Focus on | Result type |
|---|---|---|---|---|---|---|---|---|
| Disaster Risk Index (DRI) | UNDP | 2004 | 1980 ~ 2000 | Country | Earthquake, tropical cyclone, flood, drought | Expected death | Mortality | Score |
| World Risk Index (WRI) | Bündnis Entwicklung Hilft & IFHV | Annual | -*a | Country | Earthquake, tsunami, cyclone, coastal flood, riverine flood, drought, sea-level rise | Exposed people | Population & Social capability | Score & Rank |
| Global Climate Risk Index (CRI) | Germanwatch | Annual (as of 2021) | 2000 ~ 2019 | Country | Storm, flood, temperature extreme, mass movement (heat and cold wave etc.) | Death & Economic loss | Population & Economy | Score & Rank |
| INFORM Risk Index | European Commission | Biannual | -*b | Country | Earthquake, tsunami, coastal flood, riverine flood, tropical cyclone, drought | Exposed people | Population & Social capability | Score |
| Natural Disaster Hotspots | World Bank & Columbia University (The Hotspots Project) | 2005 | 1981 ~ 2000 | 2.5' grid | Earthquake, landslide, cyclone, flood, drought, volcano | Death & Economic loss | Population & Economy | Decile |
| Total Risk Index (TRI) | Beijing Normal University (World Atlas of Natural Disaster Risk) | 2015 | 1951 ~ 2013 & 1949 ~ 2009*c | Country & 0.5° grid & Comparable-geographic unit | Earthquake, landslide, tropical cyclone, flood, drought, volcano, wildfire, storm surge, sand-dust storm, heat wave, cold wave | Death & Economic loss (historical) | Population & Economy | Quintile & Decile |
| Multi-hazard Risk Index (MhRI) | | | | | | Death & Economic loss (expected) | | |

*a: WRI is calculated based on the latest available data, and its statistic years depends on the update of data sources (Frege et al., 2023);

*b: INFORM uses different sources of statistic data for different natural hazards, and the statistic years are not the same (Marin-Ferrer et al., 2017);

*c: Both TRI and MhRI use EM-DAT data from 1951 to 2013 and China Catastrophe Statistics data from 1949 to 2009 (Shi and Kasperson, 2015; Delforge et al., 2025).





## 2 Global risk assessment

### 2.1 Acquisition and selection of disaster events

The Emergency Events Database (EM-DAT) serves as the primary source of disaster records for this research. A joint initiative of the Centre for Research on the Epidemiology of Disasters (CRED) and the World Health Organization (WHO), EM-DAT was established in 1988 and has since been managed by the University of Leuven (UNDP, 2004; Delforge et al., 2025). The database documents over 26,000 significant events worldwide from 1900 to the present (Delforge et al., 2025), and its combination of open access format and authoritative sources has made it a cornerstone resource in disaster risk research (Wen et al., 2023; Mokhtari et al., 2023; Peduzzi et al., 2012; Formetta and Feyen, 2019).

Considering the timeliness of disaster statistics and reliability of historical records, this paper focuses on disaster events for the period 2013-2023 as a preliminary research object. Since EM-DAT typically verifies data from a given year at beginning of the following year (Delforge et al., 2025), the reliability of 2024 data has not been confirmed and is therefore excluded. The research prioritizes disaster categories associated with structural damage, as building destruction accounts for the majority of casualties and losses in natural hazards (Ceferino et al., 2018a; Ceferino et al., 2018b; Ceferino et al., 2024). EM-DAT's focus on significant events, those likely to trigger humanitarian crises and attract international attention, enhances the reliability of these records (Delforge et al., 2025; Mokhtari et al., 2023; Peduzzi et al., 2012; Formetta and Feyen, 2019). Using a threshold of 50 fatalities (Total Deaths), this study identifies 344 major disaster events for detailed statistical analysis.

The selection of "Total Deaths" as the primary criterion reflects its superior reliability and objectivity compared to other metrics. Population-based indicators in EM-DAT, such as "No. Affected" and "No. Homeless", are less reliable due to varying definitions and estimation methods, which often depend heavily on other metrics or overall event assessments (UNDP, 2004; Newman and Noy, 2023; Peduzzi et al., 2009). Similarly, economic indicators like "Total Damage" introduce significant statistical bias due to varying socio-economic capacities across countries with different development levels (UNDP, 2004; Peduzzi et al., 2009).

Table 2: Categories of disasters and number of recorded events.

| Disaster subgroup | Disaster type | Disaster subtype | Target disaster events | Major disaster events |
|---|---|---|---|---|
| Geophysical | Earthquake | Ground movement | 284 | 33 |
| | | Tsunami | 4 | 1 |
| Meteorological | Storm | Blizzard/Winter storm | 98 | 7 |
| | | Derecho | 6 | 0 |
| | | Extra-tropical storm | 80 | 0 |
| | | Storm (General) | 85 | 6 |
| | | Storm surge | 8 | 0 |
| | | Tornado | 56 | 3 |
| | | Tropical cyclone | 605 | 62 |
| Hydrological | Flood | Coastal flood | 3 | 0 |
| | | Flash flood | 346 | 38 |
| | | Flood (General) | 1076 | 114 |
| | | Riverine flood | 395 | 48 |
| | Mass movement | Landslide | 152 | 27 |
| | | Mudslide | 19 | 5 |
| | | **Total** | **3217** | **344** |





Table 2 presents the selected disaster categories and their corresponding event counts, following EM-DAT's classification system of subgroups, types, and subtypes. A notable aspect of this classification involves "Mass movement" and its

subtype "Landslide", which are categorized as either dry or wet events under the "Geophysical" and "Hydrological" subgroups, respectively (Delforge et al., 2025). For statistical clarity, dry landslides, representing only 1.3% of cases, are consolidated within the "Hydrological" subgroup.

This study primarily focuses on four types of natural hazards: earthquakes, storms, floods, and mass movements, encompassing three disaster subgroups and fifteen disaster subtypes. Notably, four subtypes (derecho, extra-tropical storm,

storm surge, and coastal flood) are excluded from final analysis, as there are no records of major events with at least 50 fatalities during the period. A simple calculation reveals that major disaster events corresponding to the four natural hazards account for 9.9%, 22.7%, 58.1%, and 9.3%, respectively. These percentages reflect approximate contributions to global damage and losses from 2013 to 2023.

## 2.2 Statistics of recent disaster events

A significant challenge in analyzing EM-DAT data arises from its non-standardized recording of disaster locations (Lindersson et al., 2020; Rosvold and Buhaug, 2021; Nohrstedt et al., 2022). While precise latitude-longitude coordinates are available for all earthquake events and select other disasters, the majority of events are documented only through textual descriptions of affected locations, such as city and village names. The variable nature of disaster centers and affected areas, often spanning extensive geographical regions, further complicates spatial analysis of these events.

This study addresses the standardization challenge through a systematic approach to location data processing. For each event, a single representative location is selected from the available textual descriptions, prioritizing sites that both lie near the affected area's center and demonstrate significant damage. The latitude-longitude coordinates for these selected locations are derived from administrative boundary data, providing an approximate but consistent method for disaster center identification. This standardized approach enables comprehensive geographic analysis of disaster events across the

global dataset.

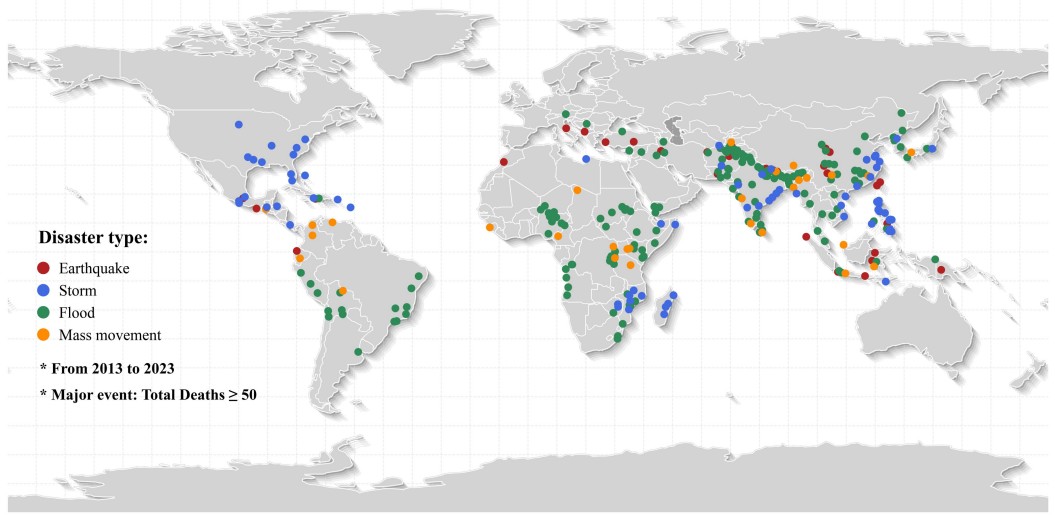

(a) Overall



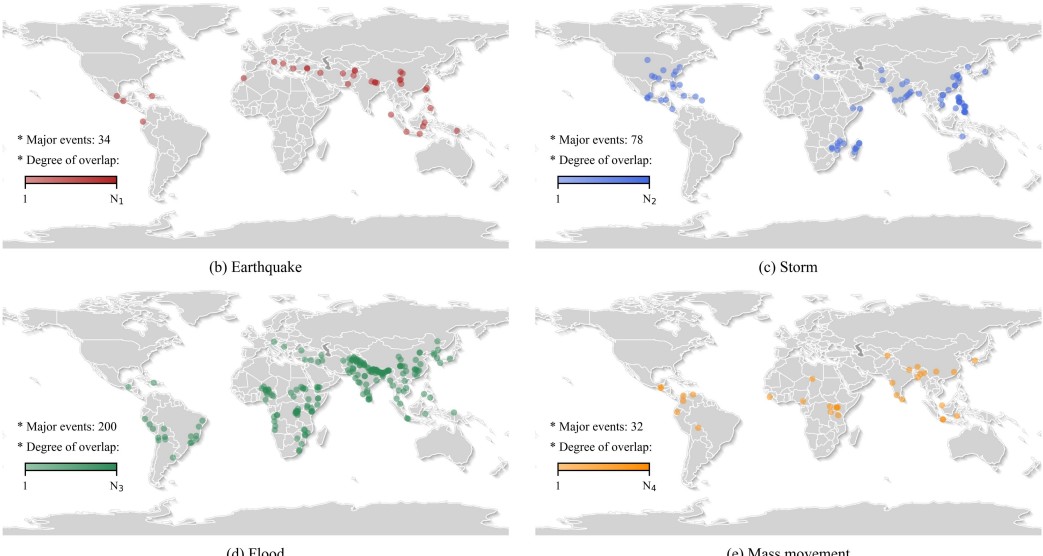

$N_1$-$N_4$: The maximum overlap degree of the data points, approximately represents the maximum frequency of major disaster events from 2013 to 2023.

**Figure 1: Geographic spatial distribution of recent disaster events.**

The resulting spatial analysis is visualized in Figure 1, with panel (a) displaying the geographic distribution of all selected disaster events plotted against a global basemap from Natural Earth (https://www.naturalearthdata.com). Panels (b) through (e) present detailed distribution and frequency patterns for different disaster types, utilizing transparency in data point visualization and separate subplots to enhance the clarity of spatial patterns. As shown in Figure 1, both individual and multiple types of natural hazards exhibit emphatic spatial clustering. This clustering is not influenced by national borders but rather demonstrates distinct geographic regional characteristics. Such patterns suggest that risk assessments based on latitude-longitude regions, rather than national boundaries, may provide a more accurate representation of disaster distribution patterns and their spatial characteristics.

Despite these natural spatial patterns, most assessment frameworks documented in Table 1 continue to use countries as their minimum assessment units. This preference reflects a practical compromise: while disaster risk depends on both regional exposure to natural hazards and socio-economic vulnerability, national-level socio-economic data offer superior accessibility and reliability compared to data for latitude-longitude regions. This raises a key research question: Can we develop a risk assessment index that harnesses the advantages of both geographic and national-level units, building on the concurrent approach used by indices such as TRI and MhRI (Shi and Kasperson, 2015)? The next section explores this methodological challenge in detail.

### 2.3 Regional risk assessments

Overall, depending on the scale of assessment, disaster risk assessments can be broadly categorized into two types: At a global or large scale, indicator-based methods utilizing selected indicators and weight allocation, as well as statistical methods grounded in historical data patterns, are typically employed (Gill and Malamud, 2014; Shi et al., 2016; Ming et al., 2015). In contrast, at smaller scales, such as in urban areas, regional disaster risk simulations often incorporate remote sensing technology (Chen et al., 2014).

### 2.3.1 World Risk Index by Bündnis Entwicklung Hilft

In disaster risk assessment, there is broad consensus that risk results from the interaction between the hazard posed by a



disaster and the vulnerability of the affected region (Koks et al., 2019; Koks et al., 2015; Little et al., 2023; Welle and Birkmann, 2015; Birkmann, 2007). The specific relationship among these three components is often represented by a multiplicative model:

$$\text{Risk} = \text{Hazard} \times \text{Vulnerability}. \tag{1}$$

This basic formula has evolved into various forms to accommodate different dimensional requirements and methodological approaches. For example, DRI expands the model to define risk as the product of hazard, population, and vulnerability (UNDP, 2004), while WRI characterizes the risk index as the geometric mean of exposure and vulnerability (Frege et al., 2023). The WRI framework, published annually since 2011 through the World Risk Report (WRR), is particularly noteworthy for its alignment with current United Nations Office for Disaster Risk Reduction (UNDRR)

terminology and its contemporary relevance (Mucke et al., 2011; Frege et al., 2023).

In WRI, exposure quantifies an area's susceptibility to various hazards, primarily considering hazard intensity and population density (Frege et al., 2023). Vulnerability, on the other hand, is composed of sensitivity, lack of coping capacities, and lack of adaptive capacities (Frege et al., 2023), closely linked to socio-economic factors. WRI is expressed as follows:

$$\text{WRI} = \sqrt{E(P, D) \times V(S, L_1, L_2)}, \tag{2}$$

where $E$ is exposure, $V$ is vulnerability, $P$ is number and share of the population regarding, $D$ is intensity levels of natural hazards, $S$ is susceptibility, $L_1$ is lack of coping capacities, and $L_2$ is lack of adaptive capacities (Frege et al., 2023).

WRR posits that the occurrence of disaster events is influenced both by severity of disaster impacts on society and vulnerability of society in responding to these impacts (Mucke et al., 2011). In other words, the risk level of disaster events

depends on both natural processes and social capabilities (Frege et al., 2023). Building on this foundation, WRR specifically emphasizes that the countries most severely affected by disaster events often lack the social capability to cope with the negative impacts of such events due to previous disasters that have eroded their resilience (Frege et al., 2023). This underscores the importance of considering the accumulative effects of recent major disaster events when conducting risk assessments. However, WRI only provides broad statistics of relevant populations from a perspective of vulnerable

groups (Frege et al., 2023), represented by two indicators, "Internally displaced persons due to natural disasters" and "Population affected by disasters in the last 5 years". Furthermore, as discussed in Section 2.1, the definitions and estimates of these population indicators may not be reliable. Moreover, since the population data of WRI are updated based on the latest global census (Frege et al., 2023), which is not refreshed annually, may lead to the neglect of recent changes in the severity of hazards, and a delay in timeliness.

**2.3.2 Accumulated Risk Index by authors**

This paper introduces the Accumulated Risk Index (ARI), a novel large-scale risk assessment framework for multiple disasters that combines the advantages of both country-level and geographic assessment units. Built upon the WRI framework while addressing its limitations, ARI specifically quantifies the accumulative effects of recent major disaster events. The expression is as follows:

$$ARI_j = \sum_{i=1}^{n_j} WRI_i \ (n_j \geq 1), \tag{3}$$

where $ARI_j$ represents the ARI of the selected region $j$, $WRI_i$ represents the WRI of the country where the $i$-th major disaster event in region $j$ took place, and $n_j$ is the number of major disaster events that have occurred recently within region $j$. When $n_j$ equals 0, it is assumed that the ARI of the region is also 0.



The selection of WRI as ARI's foundational framework is supported by four key considerations: 1) WRI maintains contemporary relevance through annual updates via WRR, ensuring continuous methodological evolution; 2) WRI's comprehensive framework of 100 indicators spans social, economic, political, and environmental dimensions (Frege et al., 2023; Mucke et al., 2022), effectively leveraging country-level data advantages; 3) WRI's exclusive focus on disaster risk encompasses diverse disaster types; 4) WRI's quantitative scoring system facilitates comparative analysis and further computational applications.

While WRI and the present study differ in their categorization of natural hazards, these differences do not significantly impact ARI's validity. WRI approaches disaster risk through a population exposure lens, measuring the proportion of people affected by hazards and emphasizing intensity and density factors (Frege et al., 2023). In contrast, this study analyzes recent major disasters through their geographic coordinates to reveal spatial distribution patterns and regional accumulative effects. These methodological differences reflect distinct but complementary approaches to risk assessment. Furthermore, WRI's scope continues to expand, as evidenced by the 2023 WRR's acknowledgment of emerging climate-related challenges and plans to incorporate new hazard types such as heat waves, cold waves, and landslides (Frege et al., 2023). This ongoing evolution suggests that the current disparities in hazard type coverage will diminish over time, justifying our treatment of these differences as negligible for the present analysis.

### 2.4 Presentation and analysis of assessment results

Specific statistical calculations for ARI depend on the selection of region $j$, which enables for the possibility of using sub-national regions as basic assessment units. However, the choice of region size is critical. An excessively large region size diminishes the advantages of using latitude-longitude regions as assessment units. Conversely, an overly small size adversely affects the results in two ways: 1) Due to the extensive impact range of large-scale hazards, which can extend over thousands of square kilometers, it becomes challenging to adequately characterize hazard types (e.g. floods and storms) that affect large areas. This limitation inhibits an effective representation of the independence of regional disaster events. 2) Relative errors will be amplified, leading to a decrease in the overall usability and reliability of data. Additionally, changes in exposure due to variations in population distribution and hazard zones over a given period will become more pronounced (Dilley, 2005), significantly affecting the timeliness of related data. Moreover, as disaster risk assessments become more refined at smaller scales (Feng et al., 2017; Smith et al., 2019), the pursuit of excessively small assessment unit sizes on a global scale clearly no longer meets practical application needs. Therefore, this study ultimately selects 5°×5° latitude-longitude grid as the minimum unit for regional division. The assessment results are also more suited for integration with remote sensing technology.

Calculation results indicate that there are 153 grids with ARI≠0, accounting for 5.9% of the total units, as shown in Figure 2 (a). It is evident from the figure that high ARI grids are primarily concentrated in Asia, while many grids have relatively low ARI levels, specifically ARI≤91 (the first quartile). To visually illustrate the differences and advantages of ARI relative to WRI, this paper converts the minimum assessment unit of WRI from countries to 5°×5° latitude-longitude grid by calculating an Equivalent WRI. Specifically, for the 153 grids with ARI≠0, countries within their respective latitude-longitude ranges are identified to obtain the corresponding WRI values. The maximum WRI among them is selected as Equivalent WRI for each grid. Equivalent WRI results obtained from the conversion are shown in Figure 2 (b). Comparing Figure 2 (a) and (b), it is evident that ARI significantly amplifies the risk disparities between different regions, particularly for high-risk areas. This disparity underscores the significant impact of the accumulative effects of multiple disasters in recent years.





**Figure 2: Global distribution of ARI and Equivalent WRI in 5°×5° latitude-longitude grids.**

To further distinguish the differences among these grids and adequately reflect the risk disparities across continental plates, major disaster events are classified into five groups according to their regional affiliations based on geographic partitioning labels from EM-DAT (Americas, Europe, Africa, Asia, and Oceania) (Delforge et al., 2025). Each group is calculated and counted independently, as shown in Figure 3. The longitude-latitude grids are ranked according to ARI statistical results, and the detailed information of each grid is shown in Table A1 in the Appendix A.



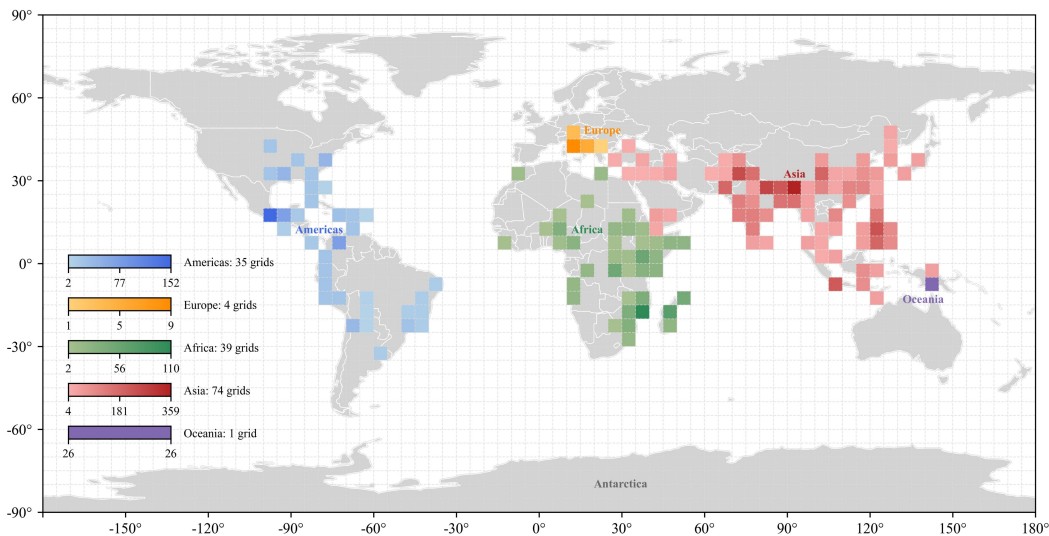

**Figure 3: Global distribution of ARI in 5°×5° latitude-longitude grids within continental plates.**

Furthermore, to specifically analyze areas with high accumulated risk of recent major disaster events that warrant increased attention, high-risk grids are defined by three times the maximum value of WRI (46.86), accounting for 9.2% of the total, as shown in Figure 4. Based on geographical coordinates, the identified high-risk grids involve ten countries: India, China, Pakistan, Philippines, Nepal, Bhutan, Indonesia, Myanmar, Bangladesh, and Mexico. All of those are developing countries, characterized by high population density, weak infrastructure development, and unequal distribution of social resources (Koks et al., 2019; Rentschler et al., 2022; Berlemann et al., 2018).

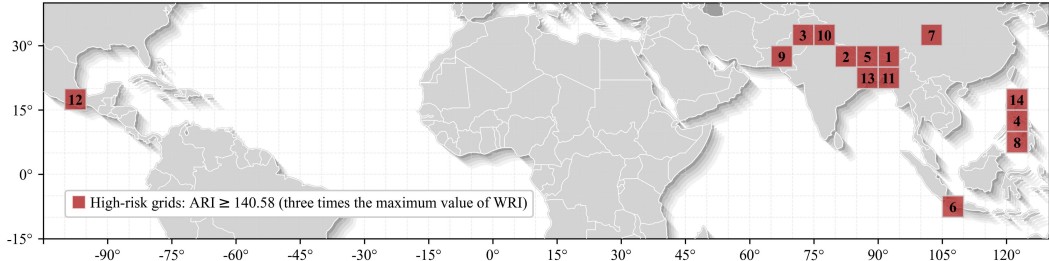

**Figure 4: Spatial distribution of high-risk grids.**

## 3 Integration and supplement of remote sensing data

ARI enables the identification of high-risk latitude-longitude grids, providing focal points for detailed risk assessment and management at finer scales. As established in Section 2.3, remote sensing technology plays a crucial role in disaster risk research at these smaller scales. However, high-resolution remote sensing data, particularly sub-meter visible spectral imaging, are primarily obtained through commercial purchases, often at significant cost. At the same time, most available open-source remote sensing data tends to focus on large cities and developed countries, overlooking a comprehensive global risk assessment. Moreover, open-source data from different sources often varies in terms of image type, size, and other attributes, further complicating integration and application. Overall, the lack of specificity and uniformity in currently available open-source remote sensing image data is a serious challenge, with high demand for data integration and supplementation.

Therefore, based on the risk assessment results in Section 2.4, this study conducts an extensive review of existing



open-source datasets, and identifies sub-meter resolution visible spectral remote sensing images for the high-risk grids, as shown in Figure 5. The data primarily come from two major sources: 1) RAMP Building Footprint Training Dataset by Replicable AI for Microplanning (RAMP) project team, in collaboration with WHO under the DevGlobal initiative, which provides relatively accurate satellite images and building digitization data for certain low- and middle-income countries or regions: https://rampml.global. 2) DigitalGlobe Open Data Program launched by the commercial remote sensing company Maxar's DigitalGlobe, offering pre- and post-event satellite images for a range of major crisis events: https://www.maxar.com/open-data.

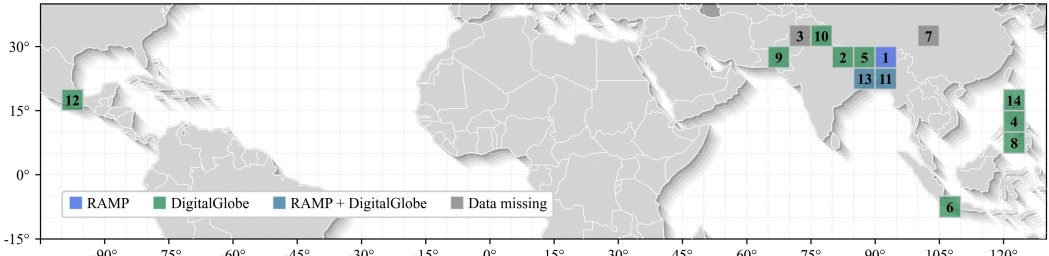

**Figure 5: Spatial distribution of existing data for high-risk grids.**

**Table 3: Detailed information of all data for high-risk grids.**

| Grid | Data source | Part name from data source | Latitude center (°) | Longitude center (°) | Resolution (m) |
|---|---|---|---|---|---|
| 1 | RAMP | Sylhet, Bangladesh | 24.96 | 91.68 | 0.27 |
| 2 | DigitalGlobe | Nepal Earthquake Nov. 2023 | 28.86 | 82.21 | 0.31 |
| **3** | **Supplement** | **-** | **33.09** | **74.79** | **0.30** |
| 4 | DigitalGlobe | Typhoon Goni | 13.18 | 123.74 | 0.33 ~ 0.53 |
| | DigitalGlobe | Typhoon Vamco | 11.61 | 122.83 | 0.34 ~ 0.50 |
| 5 | DigitalGlobe | North India Floods | 27.55 | 88.40 | 0.31 |
| 6 | DigitalGlobe | Sunda Strait Tsunami | -6.01 | 105.80 | 0.49 |
| | DigitalGlobe | Indonesia Earthquake 2022 | -6.87 | 107.06 | 0.31 |
| **7** | **Supplement** | **-** | **30.46** | **104.02** | **0.30** |
| 8 | DigitalGlobe | Typhoon Vamco | 11.61 | 122.83 | 0.34 ~ 0.50 |
| 9 | DigitalGlobe | Pakistan Flooding | 30.15 | 70.08 | 0.31 |
| | DigitalGlobe | Pakistan Earthquake | 30.12 | 67.98 | 0.45 ~ 0.55 |
| 10 | DigitalGlobe | Uttarakhand-Flooding | 30.63 | 79.63 | 0.45 ~ 0.51 |
| 11 | RAMP | Sylhet, Bangladesh | 24.96 | 91.68 | 0.27 |
| | RAMP | Chittagong, Bangladesh | 21.24 | 92.18 | 0.40 |
| | RAMP | Cox's Bazar, Bangladesh | 21.43 | 91.99 | 0.35 |
| | RAMP | Dhaka, Bangladesh | 23.81 | 90.41 | 0.30 |
| | RAMP | Barishal, Bangladesh | 22.71 | 90.35 | 0.38 |
| | DigitalGlobe | Bay of Bengal Cyclone Mocha 2023 | 20.82 | 92.40 | 0.31 |
| | DigitalGlobe | COVID19 Bangladesh | 22.38 | 91.30 | 0.29 ~ 0.54 |
| | DigitalGlobe | Cyclone Fani | 21.60 | 87.57 | 0.28 ~ 0.54 |
| 12 | DigitalGlobe | Southern Mexico Earthquake | 16.63 | -94.14 | 0.42 ~ 0.50 |
| 13 | RAMP | Jashore, Bangladesh | 23.15 | 89.16 | 0.34 |
| | DigitalGlobe | Cyclone Fani | 21.60 | 87.57 | 0.28 ~ 0.54 |
| 14 | DigitalGlobe | Typhoon Vongfong | 15.79 | 121.40 | 0.29 ~ 0.48 |
| | DigitalGlobe | Super Typhoon Mangkhut | 17.87 | 121.43 | 0.28 ~ 0.53 |



It is evident that existing open-source datasets lack relevant data for grids 3 and 7. To address this gap, this study supplements sub-meter visible spectral remote sensing images for these two grids. Based on the quantity of existing open-source data for other high-risk grids, this study supplements satellite imaging data covering a rectangular area of 50 square kilometers for grids 3 and 7 respectively, which is 100 square kilometers in total. The center point of each data area is based on the latitude-longitude coordinate of the major disaster event with the maximum "Total Deaths" in that grid. Detailed information on all data for high-risk grids is presented in Table 3. On this basis, the open-source and supplementary data are cropped and divided, then integrated by high-risk grid after removing damaged or irrelevant images, with all images standardized to 256×256 pixels. The final data compilation for each high-risk grid is summarized in Table A2 in the Appendix A.

Ultimately, the normalized data include a total of 7,583,094 TIF format images each sized 256×256. All of these are sub-meter visible spectral images with a spatial resolution of less than 0.55 m (0.27~0.54 m). This batch of data is stored in 14 grids and can be downloaded for free by grid area. The authors named this batch of data as "3H" Dataset, which means high spatial resolution remote sensing imagery within latitude-longitude grids of high accumulated risk caused by disaster events with high mortality rates. The detailed information of data from the 3H Dataset is shown in Figure 6 as an example.

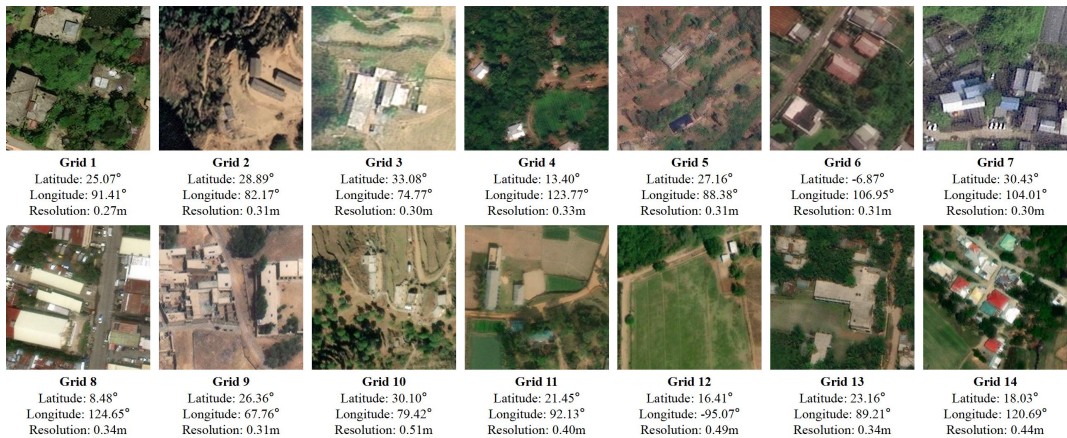

**Figure 6: Examples of data in 3H Dataset. Most of the materials are collected and prepared from RAMP and DigitalGlobe, while the remaining data are acquired and produced by the authors.**

## 4 Conclusions

This paper advances the field of disaster risk assessment by combining country-level and geographic grid analyses while emphasizing recent accumulated disaster impacts and supporting remote sensing applications. The novel Accumulated Risk Index (ARI) defined in this paper is derived from an in-depth examination of major disaster events. The resulting spatial analysis reveals emphatic clustering of both individual and multiple natural hazards, demonstrating distinct geographic regional characteristics. This phenomenon is particularly evident in regions near the Himalayas, reflecting a significant coupling effect and cascading relationship between different natural hazards in a specific geographical context.

Moreover, the calculation results of ARI indicate that high ARI grids are predominantly concentrated in Asia, particularly in developing countries, while many other grids exhibit relatively low ARI levels. Meanwhile, a comparison of ARI and Equivalent WRI reveals that ARI significantly amplifies the risk disparities between different regions, especially in high-risk areas. These findings suggest that traditional risk assessment methods may tend to underestimate the risks in high-risk areas while overestimating those in low-risk areas, leading to assessment outcomes biased toward "moderate" risk levels. This bias becomes particularly evident when incorporating the concept of regional accumulative effects. Such a





tendency is counterproductive to the rational planning and equitable allocation of global humanitarian resources, and may fail to adequately raise awareness among local governments in high-risk areas, particularly in developing countries, about the severe accumulated risks posed by multiple natural hazards.

Furthermore, open-source 3H Dataset is established in this paper based on the identification of high-risk girds by ARI. The normalized data in 3H Dataset involve ten developing countries, which are expected to effectively mitigate challenges faced by these countries in conducting disaster risk assessment and management, including the scarcity of available high-quality data, the prohibitive costs of data acquisition, and the limited access to data sources. In parallel, this open-source dataset aims to encourage greater scholarly attention to disaster risk research in developing countries and to promote the application of remote sensing technology in related fields.

## Appendix A

**Table A1: Detailed information of 153 grids (ARI≠0).**

| ARI ranking | Latitude range (°) | Longitude range (°) | Region | ARI | Equivalent WRI |
|---|---|---|---|---|---|
| 1 | 25 ~ 30 | 90 ~ 95 | Asia | 359.26 | 41.52 |
| 2 | 25 ~ 30 | 80 ~ 85 | Asia | 269.68 | 41.52 |
| 3 | 30 ~ 35 | 70 ~ 75 | Asia | 253.12 | 41.52 |
| 4 | 10 ~ 15 | 120 ~ 125 | Asia | 234.3 | 46.86 |
| 5 | 25 ~ 30 | 85 ~ 90 | Asia | 224.42 | 41.52 |
| 6 | -10 ~ -5 | 105 ~ 110 | Asia | 217.5 | 43.5 |
| 7 | 30 ~ 35 | 100 ~ 105 | Asia | 189.7 | 27.1 |
| 8 | 5 ~ 10 | 120 ~ 125 | Asia | 187.44 | 46.86 |
| 9 | 25 ~ 30 | 65 ~ 70 | Asia | 185.15 | 41.52 |
| 10 | 30 ~ 35 | 75 ~ 80 | Asia | 166.08 | 41.52 |
| 11 | 20 ~ 25 | 90 ~ 95 | Asia | 159.55 | 41.52 |
| 12 | 15 ~ 20 | -100 ~ -95 | Americas | 152.68 | 38.17 |
| 13 | 20 ~ 25 | 85 ~ 90 | Asia | 151.85 | 41.52 |
| 14 | 15 ~ 20 | 120 ~ 125 | Asia | 140.58 | 46.86 |
| 15 | 10 ~ 15 | 75 ~ 80 | Asia | 124.56 | 41.52 |
| 16 | 15 ~ 20 | 70 ~ 75 | Asia | 124.56 | 41.52 |
| 17 | 15 ~ 20 | 75 ~ 80 | Asia | 124.56 | 41.52 |
| 18 | 20 ~ 25 | 70 ~ 75 | Asia | 124.56 | 41.52 |
| 19 | -20 ~ -15 | 35 ~ 40 | Africa | 110.17 | 34.61 |
| 20 | 25 ~ 30 | 100 ~ 105 | Asia | 108.4 | 27.1 |
| 21 | 25 ~ 30 | 110 ~ 115 | Asia | 108.4 | 27.1 |
| 22 | 25 ~ 30 | 115 ~ 120 | Asia | 108.4 | 27.1 |
| 23 | 15 ~ 20 | 105 ~ 110 | Asia | 102.98 | 27.1 |
| 24 | 10 ~ 15 | 125 ~ 130 | Asia | 93.72 | 46.86 |
| 25 | 5 ~ 10 | 125 ~ 130 | Asia | 93.72 | 46.86 |
| 26 | -10 ~ -5 | 115 ~ 120 | Asia | 87 | 43.5 |
| 27 | -5 ~ 0 | 115 ~ 120 | Asia | 87 | 43.5 |
| 28 | 15 ~ 20 | 80 ~ 85 | Asia | 83.04 | 41.52 |
| 29 | 20 ~ 25 | 75 ~ 80 | Asia | 83.04 | 41.52 |
| 30 | 20 ~ 25 | 110 ~ 115 | Asia | 81.3 | 27.1 |
| 31 | 30 ~ 35 | 115 ~ 120 | Asia | 81.3 | 27.1 |
| 32 | 35 ~ 40 | 70 ~ 75 | Asia | 80.03 | 27.1 |



| 33 | 5 ~ 10 | -75 ~ -70 | Americas | 75.28 | 37.64 |
|---|---|---|---|---|---|
| 34 | 15 ~ 20 | -95 ~ -90 | Americas | 73.3 | 38.17 |
| 35 | -20 ~ -15 | 45 ~ 50 | Africa | 70.77 | 23.59 |
| 36 | 0 ~ 5 | 35 ~ 40 | Africa | 54.84 | 13.71 |
| 37 | 20 ~ 25 | 120 ~ 125 | Asia | 54.2 | 27.1 |
| 38 | 30 ~ 35 | 120 ~ 125 | Asia | 54.2 | 27.1 |
| 39 | 35 ~ 40 | 100 ~ 105 | Asia | 54.2 | 27.1 |
| 40 | 35 ~ 40 | 115 ~ 120 | Asia | 54.2 | 27.1 |
| 41 | -5 ~ 0 | 25 ~ 30 | Africa | 53.05 | 16.08 |
| 42 | 35 ~ 40 | 125 ~ 130 | Asia | 49.09 | 12.75 |
| 43 | 15 ~ 20 | 40 ~ 45 | Asia | 48.78 | 24.39 |
| 44 | -15 ~ -10 | 50 ~ 55 | Africa | 47.18 | 23.59 |
| 45 | 5 ~ 10 | 115 ~ 120 | Asia | 46.86 | 46.86 |
| 46 | 10 ~ 15 | 115 ~ 120 | Asia | 46.86 | 46.86 |
| 47 | 30 ~ 35 | -95 ~ -90 | Americas | 44.94 | 22.47 |
| 48 | 35 ~ 40 | -80 ~ -75 | Americas | 44.94 | 22.47 |
| 49 | -15 ~ -10 | 120 ~ 125 | Asia | 43.5 | 43.5 |
| 50 | -5 ~ 0 | 120 ~ 125 | Asia | 43.5 | 43.5 |
| 51 | -5 ~ 0 | 140 ~ 145 | Asia | 43.5 | 43.5 |
| 52 | 0 ~ 5 | 105 ~ 110 | Asia | 43.5 | 43.5 |
| 53 | 5 ~ 10 | 95 ~ 100 | Asia | 43.5 | 43.5 |
| 54 | 30 ~ 35 | 130 ~ 135 | Asia | 41.72 | 20.86 |
| 55 | 35 ~ 40 | 135 ~ 140 | Asia | 41.72 | 20.86 |
| 56 | 5 ~ 10 | 75 ~ 80 | Asia | 41.52 | 41.52 |
| 57 | 25 ~ 30 | 75 ~ 80 | Asia | 41.52 | 41.52 |
| 58 | -15 ~ -10 | 35 ~ 40 | Africa | 40.95 | 34.61 |
| 59 | -20 ~ -15 | 30 ~ 35 | Africa | 39.65 | 34.61 |
| 60 | -25 ~ -20 | -70 ~ -65 | Americas | 39.61 | 14.88 |
| 61 | 30 ~ 35 | 45 ~ 50 | Asia | 39.44 | 19.72 |
| 62 | 15 ~ 20 | 95 ~ 100 | Asia | 36.16 | 36.16 |
| 63 | 20 ~ 25 | 95 ~ 100 | Asia | 36.16 | 36.16 |
| 64 | 25 ~ 30 | 95 ~ 100 | Asia | 36.16 | 41.52 |
| 65 | 10 ~ 15 | 5 ~ 10 | Africa | 36.15 | 9.17 |
| 66 | 10 ~ 15 | 105 ~ 110 | Asia | 35.37 | 24.39 |
| 67 | -25 ~ -20 | 30 ~ 35 | Africa | 34.61 | 34.61 |
| 68 | -5 ~ 0 | 35 ~ 40 | Africa | 29.79 | 16.08 |
| 69 | 5 ~ 10 | 80 ~ 85 | Asia | 29.6 | 5.92 |
| 70 | 5 ~ 10 | 10 ~ 15 | Africa | 29.49 | 11.15 |
| 71 | 15 ~ 20 | -75 ~ -70 | Americas | 29.01 | 12.92 |
| 72 | 35 ~ 40 | 35 ~ 40 | Asia | 28.41 | 16.17 |
| 73 | -5 ~ 0 | 40 ~ 45 | Africa | 27.42 | 25.09 |
| 74 | 25 ~ 30 | 105 ~ 110 | Asia | 27.1 | 27.1 |
| 75 | 25 ~ 30 | 120 ~ 125 | Asia | 27.1 | 27.1 |
| 76 | 30 ~ 35 | 105 ~ 110 | Asia | 27.1 | 27.1 |
| 77 | 30 ~ 35 | 110 ~ 115 | Asia | 27.1 | 27.1 |
| 78 | 45 ~ 50 | 125 ~ 130 | Asia | 27.1 | 28.2 |
| 79 | -25 ~ -20 | -50 ~ -45 | Americas | 26.94 | 13.47 |





| 80 | -10 ~ -5 | 140 ~ 145 | Oceania | 26.3 | 43.5 |
|---|---|---|---|---|---|
| 81 | -15 ~ -10 | -80 ~ -75 | Americas | 25.55 | 25.55 |
| 82 | -15 ~ -10 | -75 ~ -70 | Americas | 25.55 | 25.55 |
| 83 | -10 ~ -5 | -80 ~ -75 | Americas | 25.55 | 25.55 |
| 84 | 0 ~ 5 | 40 ~ 45 | Africa | 25.09 | 25.09 |
| 85 | 5 ~ 10 | 45 ~ 50 | Africa | 25.09 | 25.09 |
| 86 | 5 ~ 10 | 50 ~ 55 | Africa | 25.09 | 25.09 |
| 87 | 10 ~ 15 | 40 ~ 45 | Asia | 24.39 | 24.39 |
| 88 | 15 ~ 20 | 45 ~ 50 | Asia | 24.39 | 24.39 |
| 89 | -25 ~ -20 | 45 ~ 50 | Africa | 23.59 | 23.59 |
| 90 | -5 ~ 0 | -80 ~ -75 | Americas | 23.58 | 37.64 |
| 91 | 0 ~ 5 | -80 ~ -75 | Americas | 23.58 | 37.64 |
| 92 | 10 ~ 15 | -70 ~ -65 | Americas | 23.47 | 23.47 |
| 93 | 15 ~ 20 | -70 ~ -65 | Americas | 22.47 | 22.47 |
| 94 | 20 ~ 25 | -85 ~ -80 | Americas | 22.47 | 7.76 |
| 95 | 30 ~ 35 | -100 ~ -95 | Americas | 22.47 | 22.47 |
| 96 | 30 ~ 35 | -85 ~ -80 | Americas | 22.47 | 22.47 |
| 97 | 35 ~ 40 | -90 ~ -85 | Americas | 22.47 | 22.47 |
| 98 | 40 ~ 45 | -100 ~ -95 | Americas | 22.47 | 22.47 |
| 99 | 25 ~ 30 | -85 ~ -80 | Americas | 22.47 | 22.47 |
| 100 | -15 ~ -10 | 10 ~ 15 | Africa | 22.08 | 11.04 |
| 101 | 5 ~ 10 | 100 ~ 105 | Asia | 21.09 | 24.39 |
| 102 | 10 ~ 15 | 100 ~ 105 | Asia | 21.09 | 24.39 |
| 103 | -10 ~ -5 | 10 ~ 15 | Africa | 20.77 | 11.04 |
| 104 | 10 ~ 15 | 25 ~ 30 | Africa | 20.54 | 10.27 |
| 105 | 10 ~ 15 | 30 ~ 35 | Africa | 20.54 | 10.27 |
| 106 | 30 ~ 35 | 65 ~ 70 | Asia | 20.1 | 26.45 |
| 107 | 35 ~ 40 | 45 ~ 50 | Asia | 19.72 | 19.72 |
| 108 | -5 ~ 0 | 15 ~ 20 | Africa | 19.46 | 9.73 |
| 109 | 5 ~ 10 | -85 ~ -80 | Americas | 18.82 | 18.82 |
| 110 | -30 ~ -25 | 30 ~ 35 | Africa | 18.28 | 34.61 |
| 111 | 15 ~ 20 | -90 ~ -85 | Americas | 16.79 | 38.17 |
| 112 | 35 ~ 40 | 25 ~ 30 | Asia | 16.17 | 16.17 |
| 113 | 40 ~ 45 | 30 ~ 35 | Asia | 16.17 | 28.2 |
| 114 | 35 ~ 40 | 65 ~ 70 | Asia | 16.08 | 4.02 |
| 115 | -35 ~ -30 | -60 ~ -55 | Americas | 14.88 | 14.88 |
| 116 | 0 ~ 5 | 100 ~ 105 | Asia | 14.04 | 43.5 |
| 117 | 30 ~ 35 | 20 ~ 25 | Africa | 13.93 | 17.76 |
| 118 | -25 ~ -20 | -45 ~ -40 | Americas | 13.47 | 13.47 |
| 119 | -20 ~ -15 | -50 ~ -45 | Americas | 13.47 | 13.47 |
| 120 | -20 ~ -15 | -45 ~ -40 | Americas | 13.47 | 13.47 |
| 121 | -15 ~ -10 | -45 ~ -40 | Americas | 13.47 | 13.47 |
| 122 | -10 ~ -5 | -40 ~ -35 | Americas | 13.47 | 13.47 |
| 123 | 40 ~ 45 | 125 ~ 130 | Asia | 12.75 | 27.1 |
| 124 | 10 ~ 15 | -95 ~ -90 | Americas | 11.71 | 38.17 |
| 125 | 10 ~ 15 | 35 ~ 40 | Africa | 10.27 | 10.27 |
| 126 | 15 ~ 20 | 30 ~ 35 | Africa | 10.27 | 10.27 |



| 127 | 40 ~ 45 | 10 ~ 15 | Europe | 9.97 | 9.97 |
| 128 | 30 ~ 35 | -10 ~ -5 | Africa | 9.96 | 9.96 |
| 129 | 0 ~ 5 | 25 ~ 30 | Africa | 9.73 | 9.73 |
| 130 | 5 ~ 10 | 40 ~ 45 | Africa | 9.7 | 25.09 |
| 131 | 30 ~ 35 | 40 ~ 45 | Asia | 9.23 | 12.24 |
| 132 | 5 ~ 10 | 5 ~ 10 | Africa | 9.17 | 11.15 |
| 133 | 0 ~ 5 | 30 ~ 35 | Africa | 8.4 | 13.71 |
| 134 | 30 ~ 35 | 60 ~ 65 | Asia | 8.04 | 19.72 |
| 135 | 40 ~ 45 | 15 ~ 20 | Europe | 6.23 | 9.97 |
| 136 | -15 ~ -10 | -65 ~ -60 | Americas | 5.96 | 13.47 |
| 137 | 5 ~ 10 | -15 ~ -10 | Africa | 5.32 | 6.86 |
| 138 | 5 ~ 10 | 35 ~ 40 | Africa | 4.85 | 13.71 |
| 139 | 25 ~ 30 | -80 ~ -75 | Americas | 4.78 | 4.78 |
| 140 | 45 ~ 50 | 10 ~ 15 | Europe | 4.3 | 9.97 |
| 141 | 5 ~ 10 | 25 ~ 30 | Africa | 4.25 | 10.27 |
| 142 | 30 ~ 35 | 30 ~ 35 | Asia | 4.02 | 17.76 |
| 143 | 30 ~ 35 | 35 ~ 40 | Asia | 4.02 | 12.24 |
| 144 | -15 ~ -10 | 30 ~ 35 | Africa | 3.17 | 34.61 |
| 145 | 15 ~ 20 | -65 ~ -60 | Americas | 3.01 | 22.47 |
| 146 | -25 ~ -20 | -65 ~ -60 | Americas | 2.98 | 14.88 |
| 147 | -20 ~ -15 | -65 ~ -60 | Americas | 2.98 | 13.47 |
| 148 | 20 ~ 25 | 15 ~ 20 | Africa | 2.9 | 13.93 |
| 149 | -5 ~ 0 | 30 ~ 35 | Africa | 2.69 | 16.08 |
| 150 | -25 ~ -20 | 25 ~ 30 | Africa | 2.52 | 9.14 |
| 151 | 10 ~ 15 | 0 ~ 5 | Africa | 2.16 | 9.17 |
| 152 | 15 ~ 20 | 5 ~ 10 | Africa | 2.16 | 9.52 |
| 153 | 40 ~ 45 | 20 ~ 25 | Europe | 1.75 | 8.58 |

**Table A2: Summary of integrated data for high-risk grids.**

| Grid | Data source | Number | Resolution (m) | Size (px) | Format |
|------|-------------|--------|----------------|-----------|--------|
| 1 | RAMP | 3,955 | 0.27 | 256 × 256 | TIF |
| 2 | DigitalGlobe | 18,569 | 0.31 | 256 × 256 | TIF |
| 3 | Supplement | 9,853 | 0.30 | 256 × 256 | TIF |
| 4 | DigitalGlobe | 634,795 | 0.33 ~ 0.53 | 256 × 256 | TIF |
| 5 | DigitalGlobe | 82,670 | 0.31 | 256 × 256 | TIF |
| 6 | DigitalGlobe | 365,809 | 0.31 ~ 0.49 | 256 × 256 | TIF |
| 7 | Supplement | 9,982 | 0.30 | 256 × 256 | TIF |
| 8 | DigitalGlobe | 58,518 | 0.34 ~ 0.49 | 256 × 256 | TIF |
| 9 | DigitalGlobe | 645,722 | 0.31 | 256 × 256 | TIF |
| 10 | DigitalGlobe | 244,489 | 0.46 ~ 0.51 | 256 × 256 | TIF |
| 11 | RAMP + DigitalGlobe | 1,894,732 | 0.27 ~ 0.54 | 256 × 256 | TIF |
| 12 | DigitalGlobe | 167,606 | 0.43 ~ 0.50 | 256 × 256 | TIF |
| 13 | RAMP + DigitalGlobe | 1,403,704 | 0.29 ~ 0.53 | 256 × 256 | TIF |
| 14 | DigitalGlobe | 2,042,690 | 0.31 ~ 0.53 | 256 × 256 | TIF |

**Data availability**

The 3H Dataset generated during the current study is scheduled for open-access publication on the Zenodo platform. The



dataset analysed during the current study is available in the [EM-DAT] repository, [https://www.emdat.be/]. The datasets used or analysed during the current study are available from the corresponding author on reasonable request.

**Author contribution**

E.Z. prepared the figures and wrote the manuscript. Q.K. prepared the figures and revised the final manuscript. All authors reviewed the manuscript.

**Competing interests**

The authors declare that they have no conflict of interest.

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
