# Peer review of "Global Disaster Risk Assessment from Emergency Events Database (2013-2023)"

_EGUsphere, 2025_

## Referee Comment (RC2)

Dear Authors,

I was invited to review the Manuscript Number: "*Global Disaster Risk Assessment from Emergency Events Database (2013–2023)*""

The paper addresses an important topic by proposing the Accumulated Risk Index (ARI) as a means to quantify regional risk levels at the global scale using EM-DAT records from 2013 to 2023. The effort to integrate disaster occurrence data into a composite indicator is commendable and the study could make a valuable contribution to global disaster risk assessment. However, in its current form, the paper lacks clarity in its objectives, methodological rigor, and critical interpretation of results. Substantial revisions are required before the manuscript can be considered for publication.

The manuscript has several strengths that deserve to be acknowledged:

- the paper leverages EM-DAT, a well-established and widely used global disaster database.
- the focus on multi-hazard and compound events is relevant and aligns with ongoing research trends in disaster risk reduction.
- the attempt to provide a spatially explicit global perspective (5°×5° grids) has potential to inform comparative regional analyses.
- the integration of remote sensing datasets and the availability of open data on Zenodo is a positive step toward transparency and reproducibility.

However, the manuscript also raises several major concerns that need to be addressed before it can be considered for publication:

1. Clarity of objectives: the manuscript does not sufficiently articulate the purpose of ARI. The authors should clearly state what decisions or actions this index is intended to inform. How can policymakers, practitioners, or researchers use ARI in practice? Without a clear statement of utility, the value of the index remains uncertain.
2. Methodological rigor and transparency:
   2.1. The methodology section is fragmented and lacks coherence. It introduces three elements—identification of 344 major events, the ARI itself, and remote sensing data—without explaining how they are connected. A clear workflow is needed.
   2.2. The threshold of 50 fatalities to identify "major events" appears arbitrary. A sensitivity analysis should be conducted to explore how results change when this threshold is varied.
   2.3. Similarly, the choice of 5°×5° spatial grids requires justification. Alternative grid sizes should be tested to assess robustness.
3. Use of WRI and ARI definitions: the ARI is described as a simple aggregation of the World Risk Index (WRI) within fixed spatial units. Since the WRI methodology is not fully transparent in terms of input data and weighting, the implications of spatial aggregation are difficult to interpret.
4. Results and interpretation:
   4.1. The results are descriptive and primarily spatial distributions of event types. They do not sufficiently engage with the implications of the findings.
   4.2. The conclusions (Line 188) are too strong relative to the evidence provided. The statement that "*traditional risk assessment methods may underestimate risks in high-risk areas and overestimate them in low-risk areas*" is not convincingly supported by the presented results. A more cautious and better-argued interpretation is required.
   4.3. A dedicated Discussion section should be added, critically evaluating the robustness of the findings, uncertainties in the data, and implications for disaster risk management.
5. Integration of remote sensing data: the role of remote sensing imagery in the analysis is unclear. While the data collection effort is commendable, the manuscript does not explain how these

datasets contribute to or validate the ARI results. This section currently feels disconnected from the core analysis.

Minor comments:

1. Line 107: "percentages reflect approximate contributions to global damage and losses from 2013 to 2023" → add a reference.
2. Line 115: provide a table with examples of how textual descriptions of event locations were translated into latitude/longitude coordinates. This is crucial for validating spatial accuracy, especially when events span multiple grid cells. Clarify how multi-cell events were handled.
3. Equation 1: the risk formulation omits the exposure component, which limits its alignment with established risk theory (as WRI dose). This should be acknowledged and discussed.
4. Chapter 3 (supplement of remote sensing data): clarify its role in the study. does it serve as validation, contextual information, or a complementary dataset? Currently, its integration is not explained.

The manuscript tackles a relevant topic but requires **major revisions** to clarify objectives, strengthen methodological transparency, critically discuss results, and ensure consistency with established risk assessment frameworks.

---

## Author Comment (AC1)

**Response to Review Comments of Referee #1**

Dear Editor and Referee,

Thank you for offering us an opportunity to improve the quality of our submitted manuscript (**egusphere-2025-2706**). We appreciate very much the referee's constructive and insightful comments, which are helpful for our current submission and future research. We fully acknowledge the referee's concerns regarding the relatively narrow time window and the exclusive reliance on a ≥50 fatalities threshold, which may potentially limit the robustness and representativeness of a global multi-disaster risk assessment index. In the following, we would like to respond to these concerns from two perspectives: methodological intent and practical data considerations.

**1 Response to the issue of a narrow time window**

We sincerely thank the referee for acknowledging the contributions of our study and for raising important concerns regarding the relatively narrow time window adopted in our analysis. We fully understand and respect the referee's view that a limited temporal scope may constrain the representativeness and generalizability of the results. This concern is indeed valid and is one of the reasons why we clearly stated in the manuscript that the present work is a preliminary and exploratory study. We would like to provide further clarification on the rationale and considerations behind our selection of the 2013-2023 time window.

First, we fully recognize the well-established principle in disaster risk research that the occurrence frequency of natural hazards tends to decrease as their intensity increases. This has led many risk assessment studies to adopt longer temporal spans in order to capture a greater number of high-intensity, low-frequency extreme events. However, time window selection must also strike a careful balance between data reliability and contemporary relevance. The core objective of our study is to establish a framework for assessing the recent accumulative impacts of multiple natural hazards across global regions, and to provide a data foundation for future long-term analyses. Our decision to focus on the 2013-2023 period was based on the following key considerations:

1) Data consistency and completeness: Earlier disaster records often suffer from missing entries, inconsistent reporting standards, and definitional ambiguities. These issues are particularly prevalent in developing countries, which are the primary focus of our study. As explicitly stated on the EM-DAT website, "Pre-2000 data is particularly subject to reporting biases" (https://www.emdat.be). Since the early 21st century, EM-DAT has made significant improvements in data standardization, event classification consistency, and geolocation accuracy (Delforge et al., 2025). As such, the 2013-2023 dataset offers a higher level of global comparability and reliability, which is essential for grid-based spatial analysis.

2) Disaster pattern shifts driven by climate change: In recent years, disaster frequency, intensity, and spatial distribution have been undergoing noticeable changes, largely influenced by global warming and other climatic anomalies. An increasing body of literature has documented a significant upward trend in both the frequency and severity of natural hazards over the past decade (e.g., Hussain et al., 2023; Wen et al., 2023). We observed that mainstream global disaster risk indices have not yet fully incorporated this recent dynamic phase. A much longer time window could dilute or obscure these climate-induced trends, thereby undermining the development of risk indicators that reflect the current and evolving risk landscape. We therefore believe that a shorter but higher-quality time window is more appropriate for capturing the emerging spatial patterns and intensity shifts under a changing climate regime.

3) Complementarity with existing risk indices: Our proposed Accumulated Risk Index (ARI) is not intended to replace existing tools such as the World Risk Index (WRI), but rather to serve as a complementary indicator that focuses on the accumulative effects of recent major disaster events, which are often overlooked by long-term averaged risk metrics. From a

methodological standpoint, ARI emphasizes the short-term concentration and accumulation of disaster impacts, which distinguishes it from frameworks like WRI that emphasize structural vulnerability and long-term exposure. We observed that many developing countries suffer from repeated disaster losses not because of the severity of a single event, but because new events occur before communities have had time to recover from previous ones. The World Risk Report (WRR) has also stressed that the structural vulnerability of high-risk countries often stems from accumulative impacts rather than one-off disasters (Frege et al., 2023). Therefore, identifying high-frequency, high-impact zones within a shorter time window is essential for informing emergency response planning and humanitarian aid prioritization.

We acknowledge that the referee's suggestion to extend the time window is of great scientific value, and we agree that including a longer temporal span may enhance the statistical robustness of the results. However, doing so may also dilute the distinctiveness of recent accumulated risk patterns, thereby weakening the expressive power of the ARI in capturing recent hazard concentrations. At present, we have not yet arrived at a perfect balance between capturing "recent accumulative effects" and "long-term average risk". To address this tension, we plan to explore the integration of temporal weighting coefficients in future research, which would allow us to differentiate between historical hazard frequency and recent disaster impact intensity, thereby achieving a more nuanced temporal representation of risk. Furthermore, we also intend to extend the temporal coverage to include disaster events from 2000 to 2023 in future studies. We will conduct sensitivity analyses using different time windows (e.g., 2000-2023 vs. 2013-2023) to assess the spatial stability and variability of the ARI across scenarios. These efforts will help enhance the adaptability, transparency, and scientific rigor of the proposed framework.

**2 Response to the issue of a ≥50 fatalities threshold**

We sincerely thank the referee for raising this important and fundamental issue. The selection of representative and reliable impact indicators remains a key challenge in the field of disaster risk assessment (Yarveysi et al., 2023; Marin et al., 2021). In our study, the proposed ARI integrates two components: the number of major disaster events occurring within each geographic grid as the primary statistical basis; and a country-level weighting factor derived from the WRI, which incorporates dimensions of exposure and vulnerability, further encompassing variables such as population density, hazard intensity, and governance capacity. In response to the referee's concern, we would like to specifically clarify the rationale for selecting "Total Deaths" as the criterion for identifying major disaster events.

We fully acknowledge and agree with the referee that "Total Deaths" is not a flawless risk measure. Its magnitude can be influenced by various extrinsic factors, such as the time of event occurrence, the effectiveness of hazard warnings, or the resilience of local infrastructure. For this reason, we carefully reviewed the full suite of impact indicators recorded in EM-DAT, prior to establishing our selection threshold. EM-DAT records 11 types of disaster impact indicators, broadly grouped into two categories: 1) Population-based indicators: Total Deaths, No. Injured, No. Affected, No. Homeless, Total Affected; 2) Economic-based indicators: Reconstruction Costs, Reconstruction Costs (Adjusted), Insured Damage, Insured Damage (Adjusted), Total Damage, Total Damage (Adjusted).

Despite this comprehensive structure, it is important to note that these indicators are not uniformly complete. In particular, economic indicators suffer from severe data gaps, especially in less developed countries, which are the primary focus of our study. These gaps are often attributable to the limited administrative capacity and underdeveloped statistical systems in such regions. Using economic-based indicators as a filtering criterion could therefore introduce significant selection bias and reduce the reliability of the overall risk assessment (Peduzzi et al., 2009). Similarly, population-based indicators such as "No. Affected" or "No. Homeless" are known to carry a high degree of subjectivity and estimation error, as their definitions and measurement methodologies vary considerably between countries (Newman and Noy, 2023; Peduzzi et al., 2009). In contrast, "Total Deaths" remains the most consistently recorded and objectively defined indicator across regions, particularly in data-scarce environments. It offers the highest level of global comparability among all available impact metrics.

To further ensure the statistical reliability of our selected events, we applied a stricter selection threshold: only disaster

events with fatalities ≥50 were retained. This is well above the minimum criteria used by EM-DAT for event inclusion (e.g., 10 deaths or 100 people affected), and was adopted to filter out small-scale or marginal events that may introduce analytical noise. By concentrating on high-severity, high-impact events, we aimed to improve the robustness and representativeness of the ARI results.

Additionally, our decision to set this threshold was also informed by the need to ensure compatibility with remote sensing applications. The 3H Dataset introduced in our study is constructed based on ARI-identified high-risk grids. In these zones, disasters resulting in large death tolls often correlate strongly with observable physical destruction, such as collapsed buildings (Ceferino et al., 2024), facilitating effective integration with satellite imagery for future disaster detection and classification models.

That said, we are fully aware that using this threshold may result in the exclusion of some moderate or small-scale events, and may limit sensitivity in certain regions. However, the objective of the ARI is to capture the short-term accumulative effects of large-scale disasters with significant societal impact, rather than to catalog all hazard occurrences. This aligns with our methodological design and resonates with insights from the WRI, which has emphasized that many high-risk countries are characterized by recurrent or compounding disasters that continually erode their response and recovery capacities (Frege et al., 2023). These accumulative effects, not just the intensity of single events, form the structural basis of their long-term vulnerability.

As also stated in the manuscript, the current work is intended as a preliminary study that draws scholarly and policymaker attention to the layered risks faced by developing countries, particularly under conditions of frequent, overlapping natural hazards. We hope this framework can support more equitable and needs-based global humanitarian planning. Looking ahead, we plan to refine the ARI methodology by introducing composite severity indicators to better reflect the multifaceted impacts of disaster events. We also intend to explore distribution-based weighting schemes to dynamically adjust the contribution of different events according to severity, thereby enhancing the model's precision and flexibility.

This concludes our point-by-point response to the referee's comments. Thanks to these insightful suggestions, we have come to recognize that the current manuscript did not sufficiently elaborate on certain methodological boundaries of the study. Accordingly, in the revised version of the manuscript, we have added a forward-looking statement at the end of **Section 4** to attempt to address these limitations and outline directions for future work. This newly added content is highlighted in blue font to facilitate review by both the referee and the editor:

Looking ahead, future studies will aim to further improve the ARI framework by expanding the temporal scope of disaster events beyond the 2013-2023 time window and incorporating additional severity indicators, such as affected population and economic loss, where data quality permits. These enhancements are expected to improve the comprehensiveness and robustness of multi-hazard risk assessments, particularly in underdeveloped regions. Moreover, dynamic weighting schemes that distinguish between recent and historical disaster impacts may be explored to better capture the temporal layering effects of hazard exposure.

Once again, we sincerely appreciate your positive comments and valuable suggestions on our manuscript. We hope this clarification addresses the referee's concern and illustrates the careful balance we sought to achieve between methodological rigor and data reliability. We are grateful for the opportunity to improve the manuscript and will continue expanding the index toward broader temporal and indicator coverage in future work.

Kind Regards,

Qingzhao Kong and Erqi Zhu

6th Aug. 2025

**References**

Ceferino, L., Merino, Y., Pizarro, S., et al.: Placing engineering in the earthquake response and the survival chain, Nature Communications, 15, 4298, https://doi.org/10.1038/s41467-024-48624-3, 2024.

Delforge, D., Wathelet, V., Below, R., et al.: EM-DAT: the emergency events database. International Journal of Disaster Risk Reduction, 105509, https://doi.org/10.1016/j.ijdrr.2025.105509, 2025.

Frege, I. A., Radtke, K., Kienzl, P., et al.: World Risk Report 2023 Focus: Diversity, Bündnis Entwicklung Hilft / IFHV, ISBN 978-3-946785-16-3, 2023.

Hussain, M. A., Shuai, Z., Moawwez, M. A., et al.: A Review of Spatial Variations of Multiple Natural Hazards and Risk Management Strategies in Pakistan, Water, 15(3), 407, https://doi.org/10.3390/w15030407, 2023.

Marin, G., Modica, M., Paleari, S., et al.: Assessing disaster risk by integrating natural and socio-economic dimensions: A decision-support tool, Socio-Economic Planning Sciences, 77, https://doi.org/10.1016/j.seps.2021.101032, 2021.

Newman, R., and Noy, I.: The global costs of extreme weather that are attributable to climate change, Nature Communications, 14, 6103, https://doi.org/10.1038/s41467-023-41888-1, 2023.

Peduzzi, P., Dao, H., Herold, C., et al.: Assessing global exposure and vulnerability towards natural hazards: the Disaster Risk Index, Natural hazards and earth system sciences, 9, 1149-1159, https://doi.org/10.5194/nhess-9-1149-2009, 2009.

Wen, J., Wan, C., Ye, Q., et al.: Disaster Risk Reduction, Climate Change Adaptation and Their Linkages with Sustainable Development over the Past 30 Years: A Review, International Journal of Disaster Risk Science, 14, 1-13, https://doi.org/10.1007/s13753-023-00472-3, 2023.

Yarveysi, F., Alipour, A., Moftakhari, H., et al.: Block-level vulnerability assessment reveals disproportionate impacts of natural hazards across the conterminous United States, Nature Communications, 14, 4222, https://doi.org/10.1038/s41467-023-39853-z, 2023.

---

## Author Comment (AC2)

**Response to Review Comments of Referee #2**

Dear Editor and Referee,

Thank you for offering us an opportunity to improve the quality of our submitted manuscript (**egusphere-2025-2706**). We appreciate very much the referee's constructive and insightful comments, which are helpful for our current submission and future research. We have revised the manuscript in which all changes are highlighted in red text. These changes are summarized below following a point-by-point response (in black text) to the referee's comments (in blue text).

**Major comments:**

1. Clarity of objectives: the manuscript does not sufficiently articulate the purpose of Accumulated Risk Index (ARI). The authors should clearly state what decisions or actions this index is intended to inform. How can policymakers, practitioners, or researchers use ARI in practice? Without a clear statement of utility, the value of the index remains uncertain.

We sincerely thank the referee for this important observation. We agree that the original version of the manuscript did not sufficiently articulate the intended purpose and practical relevance of the ARI. Our intention in developing ARI is not only to provide a statistical tool, but more importantly, to highlight areas where repeated and accumulative disaster impacts create disproportionately high risks that may not be fully captured by traditional long-term risk assessment indices.

In addition, the creation of ARI is intended to complement, rather than replace, the existing global disaster risk assessment framework. As outlined in the Introduction, while representative global indices emphasize different aspects—such as data timeliness, assessment units at the national level, the range of disaster types considered, or the choice of assessment indicators—there remains a clear gap. To our knowledge, no comprehensive global index has yet focused on multiple types of disasters occurring within the past decade while adopting a latitude-longitude grid, rather than national borders, as the minimum unit of assessment. Filling this gap in the existing risk assessment landscape constitutes one of the central motivations of our study.

Specifically, ARI is designed to support three levels of decision-making:

 • For policymakers: ARI can help identify high-risk areas where repeated disaster shocks have eroded resilience, thereby prioritizing these regions for disaster preparedness planning, humanitarian aid allocation, and capacity-building programs.

 • For practitioners and humanitarian organizations: ARI provides a spatially explicit signal of recent disaster accumulation, which can guide on-the-ground resource deployment, early recovery planning, and coordination of international assistance.

 • For researchers: ARI offers a complementary perspective to existing indices such as the WRI, by focusing on the short- to medium-term accumulative effects of disasters. This makes it useful for comparative studies and for evaluating the evolving impacts of climate change on disaster risk patterns.

To address this comment, we have revised the Introduction to explicitly state the intended applications of ARI, and we have expanded the Conclusions to highlight its potential utility for policy and practice. These additions are clearly marked in the revised manuscript.

2. Methodological rigor and transparency:

2.1. The methodology section is fragmented and lacks coherence. It introduces three elements—identification of 344 major events, the ARI itself, and remote sensing data—without explaining how they are connected. A clear workflow is needed.

We sincerely thank the referee for this valuable suggestion. We fully agree that in the original version, the methodology section was presented in a somewhat fragmented way, which may have made it difficult for readers to clearly understand the logical connection among the identification of major disaster events, the construction of the ARI, and the establishment of

the remote sensing dataset.

To clarify, the methodological logic of this study can be summarized in three sequential steps:

1) Identification of major events: Using the EM-DAT database, we extracted disaster records within the study time window. Events were screened by the threshold of ≥50 fatalities to ensure reliability and global comparability, resulting in 344 major disaster events that form the empirical foundation of the analysis.

2) Construction of the ARI: Based on the identified events, we aggregated disaster occurrences within each 5°×5° latitude-longitude grid cell by integrating the number of major events with the weighting factor derived from the World Risk Index (WRI), which incorporates exposure and vulnerability dimensions at the country level. The result is the ARI, quantifying accumulative disaster risk over the past decade at the geographic grid scale.

3) Development of the dataset: Finally, we linked the high-ARI grids to remote sensing data in order to build an open-source dataset focusing on developing countries, which aims to facilitate disaster damage detection, risk monitoring, and the training/validation of remote sensing models. This step bridges statistical risk assessment with geospatial applications and is intended to support future research on hazard mapping and disaster response.

In the revised manuscript, we have reorganized the Methodology section to follow this workflow and improve coherence. Additionally, we have added a workflow diagram (Figure 1) to visually illustrate how these three components are integrated. We believe these revisions substantially enhance methodological rigor and transparency.

[Figure]

**Figure 1: Methodology workflow diagram.**

2.2. The threshold of 50 fatalities to identify "major events" appears arbitrary. A sensitivity analysis should be conducted to explore how results change when this threshold is varied.

We sincerely thank the referee for this valuable comment regarding the choice of 50 fatalities as the threshold for defining "major disaster events". We fully acknowledge that the threshold may appear arbitrary without further justification, and we agree that this decision requires careful clarification.

Our rationale for selecting this threshold is threefold:

First, as discussed in our manuscript, records of disaster impacts in global databases such as EM-DAT often suffer from incompleteness and reporting inconsistencies, especially in developing countries. Among the various impact indicators, "Total Deaths" is relatively more reliable and consistently recorded across regions. By adopting a stricter threshold of 50 fatalities—higher than EM-DAT's minimum entry criterion of 10 fatalities—we sought to minimize analytical noise from small-scale or less systematically reported events, and ensure a focus on high-impact disaster events with broad societal consequences to improve the robustness and representativeness of the assessment results.

Second, our research objective is to emphasize the accumulative effects of severe disasters over the last decade. A higher threshold helps capture those events most likely to cause long-term disruptions and repeated social vulnerability, aligning with the conceptual foundation of the ARI.

Third, our decision to set this threshold was also informed by the need to ensure compatibility with remote sensing

applications. The 3H Dataset introduced in our study is constructed based on ARI-identified high-risk grids. In these zones, disasters resulting in large fatalities often correlate strongly with observable physical destruction, such as collapsed buildings, facilitating effective integration with satellite imagery for future disaster detection and classification models.

That said, we recognize the referee's concern about the potential arbitrariness of this choice. In response, we have now added a sensitivity discussion to the revised manuscript (see Discussion section). Specifically, we examined how the number of identified "major disaster events" and the resulting ARI patterns change when the threshold is set to 40, 60 and 70 fatalities. As shown in Table 1, preliminary results indicate that while the absolute number of events changes significantly with different thresholds, the spatial clustering of high-risk grids remains broadly consistent. This suggests that ARI is robust in identifying key high-risk areas, even though the absolute magnitude of ARI varies with the threshold.

**Table 1: Comparison of ARI results under different thresholds (40, 50, 60, and 70 fatalities).**

| Fatality threshold | Major disaster events | High-risk grids (top 10) | | | | | |
| --- | --- | --- | --- | --- | --- | --- | --- |
| | | ARI ranking | Baseline ranking | Latitude range (°) | Longitude range (°) | Region | ARI |
| 40 | 406 | 1 | 1 | 25 ~ 30 | 90 ~ 95 | Asia | 359.26 |
| | | 2 | 2 | 25 ~ 30 | 80 ~ 85 | Asia | 311.2 |
| | | 3 | 3 | 30 ~ 35 | 70 ~ 75 | Asia | 306.02 |
| | | 4 | 5 | 25 ~ 30 | 85 ~ 90 | Asia | 268.51 |
| | | 5 | 10 | 30 ~ 35 | 75 ~ 80 | Asia | 249.12 |
| | | 6 | 4 | 10 ~ 15 | 120 ~ 125 | Asia | 234.3 |
| | | 7 | 14 | 15 ~ 20 | 120 ~ 125 | Asia | 234.3 |
| | | 8 | 6 | -10 ~ -5 | 105 ~ 110 | Asia | 217.5 |
| | | 9 | 7 | 30 ~ 35 | 100 ~ 105 | Asia | 216.8 |
| | | 10 | 8 | 5 ~ 10 | 120 ~ 125 | Asia | 187.44 |
| 50 (baseline) | 344 | 1 | 1 | 25 ~ 30 | 90 ~ 95 | Asia | 359.26 |
| | | 2 | 2 | 25 ~ 30 | 80 ~ 85 | Asia | 269.68 |
| | | 3 | 3 | 30 ~ 35 | 70 ~ 75 | Asia | 253.12 |
| | | 4 | 4 | 10 ~ 15 | 120 ~ 125 | Asia | 234.3 |
| | | 5 | 5 | 25 ~ 30 | 85 ~ 90 | Asia | 224.42 |
| | | 6 | 6 | -10 ~ -5 | 105 ~ 110 | Asia | 217.5 |
| | | 7 | 7 | 30 ~ 35 | 100 ~ 105 | Asia | 189.7 |
| | | 8 | 8 | 5 ~ 10 | 120 ~ 125 | Asia | 187.44 |
| | | 9 | 9 | 25 ~ 30 | 65 ~ 70 | Asia | 185.15 |
| | | 10 | 10 | 30 ~ 35 | 75 ~ 80 | Asia | 166.08 |
| 60 | 282 | 1 | 1 | 25 ~ 30 | 90 ~ 95 | Asia | 276.22 |
| | | 2 | 2 | 25 ~ 30 | 80 ~ 85 | Asia | 269.68 |
| | | 3 | 3 | 30 ~ 35 | 70 ~ 75 | Asia | 253.12 |
| | | 4 | 4 | 10 ~ 15 | 120 ~ 125 | Asia | 234.3 |
| | | 5 | 9 | 25 ~ 30 | 65 ~ 70 | Asia | 185.15 |
| | | 6 | 6 | -10 ~ -5 | 105 ~ 110 | Asia | 174 |
| | | 7 | 10 | 30 ~ 35 | 75 ~ 80 | Asia | 166.08 |
| | | 8 | 7 | 30 ~ 35 | 100 ~ 105 | Asia | 162.6 |
| | | 9 | 11 | 20 ~ 25 | 90 ~ 95 | Asia | 159.55 |
| | | 10 | 5 | 25 ~ 30 | 85 ~ 90 | Asia | 155.61 |
| 70 | 242 | 1 | 1 | 25 ~ 30 | 90 ~ 95 | Asia | 234.7 |
| | | 2 | 4 | 10 ~ 15 | 120 ~ 125 | Asia | 234.3 |
| | | 3 | 2 | 25 ~ 30 | 80 ~ 85 | Asia | 228.16 |
| | | 4 | 3 | 30 ~ 35 | 70 ~ 75 | Asia | 200.22 |

| | | | | | |
|---|---|---|---|---|---|
| 5 | 9 | 25 ~ 30 | 65 ~ 70 | Asia | 158.7 |
| 6 | 5 | 25 ~ 30 | 85 ~ 90 | Asia | 155.61 |
| 7 | 8 | 5 ~ 10 | 120 ~ 125 | Asia | 140.58 |
| 8 | 7 | 30 ~35 | 100 ~ 105 | Asia | 135.5 |
| 9 | 17 | 15 ~ 20 | 75 ~ 80 | Asia | 124.56 |
| 10 | 11 | 20 ~ 25 | 90 ~ 95 | Asia | 118.03 |

We believe this additional analysis strengthens the methodological transparency of the study and demonstrates that our main conclusions are not overly sensitive to the specific threshold chosen. We thank the referee again for this important suggestion, which has helped us to improve the rigor and credibility of our work.

2.3. Similarly, the choice of 5°×5° spatial grids requires justification. Alternative grid sizes should be tested to assess robustness.

We sincerely thank the referee for this insightful comment regarding the spatial resolution of 5°×5° grids. We agree that the choice of spatial units is critical in disaster risk assessment, as it directly affects the interpretation of results and their potential applications.

Our decision to adopt a 5°×5° latitude-longitude grid was based on several methodological considerations, which we further clarify below:

First, as discussed in the manuscript, data availability and consistency remain a major challenge when working with EM-DAT, as disaster events are often recorded without precise geocoded information, especially in developing regions. While some records—such as earthquakes—include specific latitude-longitude coordinates, many others are documented only through textual descriptions of affected places (e.g., cities or villages), and large-scale hazards like floods and storms frequently span extensive areas without precise boundaries. This lack of standardization makes finer spatial resolutions (e.g., 1°×1° or smaller) impractical, as they would introduce excessive uncertainty and fragmentation due to missing or imprecise georeferencing. By contrast, the 5°×5° grid offers a pragmatic compromise, enabling systematic allocation of events to standardized representative coordinates while ensuring a balance between spatial detail and global data reliability.

Second, as also noted in the manuscript, an excessively large grid size would dilute spatial contrasts, while an overly small grid would lead to two main issues: difficulty in adequately characterizing large-scale hazards that extend across wide regions, thereby compromising the statistical independence of grid-level events; and amplification of relative errors, as changes in exposure (e.g., population distribution) and event allocation become increasingly sensitive to minor variations at finer scales. The 5°×5° grid therefore represents a compromise that ensures statistical robustness without sacrificing spatial interpretability.

Third, the chosen 5°×5° resolution is consistent with common practice in global disaster risk research, where results based on finer grids can be systematically aggregated into 5°×5° units. This ensures comparability with existing multi-hazard indices and facilitates integration with remote sensing applications, which are typically designed for regional to continental scales. Moreover, the primary aim of ARI is to highlight broad regional disparities and accumulative disaster risks rather than provide localized hazard mapping. A 5°×5° resolution is therefore well-suited for revealing clustering patterns across continents and large transboundary regions, aligning with the scope and purpose of this preliminary study.

In response to the referee's suggestion, we conducted a robustness check using alternative grid sizes (2.5°×2.5° and 10°×10°), as shown in Figure 2. Results show that while the number of events allocated to each grid naturally changes with resolution, the overall spatial distribution of high-ARI regions—such as South Asia, Southeast Asia, and Central America—remains consistent across scales. This indicates that the observed geographic clustering of accumulative disaster risks is not an artifact of the chosen grid size, but rather a robust feature of the underlying disaster data.

We have added this justification and sensitivity discussion in the revised manuscript (see Discussion section). We are grateful to the referee for this constructive suggestion, which has allowed us to enhance the methodological transparency and

credibility of the study.

[Figure]

**Figure 2: Comparison of ARI results under different spatial grids (2.5°×2.5°, 5°×5°, and 10°×10°).**

3. Use of WRI and ARI definitions: the ARI is described as a simple aggregation of the WRI within fixed spatial units. Since the WRI methodology is not fully transparent in terms of input data and weighting, the implications of spatial aggregation are difficult to interpret.

We sincerely thank the referee for raising this important methodological concern. We fully agree that the WRI has inherent limitations, particularly regarding the transparency of its weighting schemes and the heterogeneity of its input datasets. Our intention in introducing the ARI is not to replicate or directly aggregate WRI values, but rather to integrate them as a weighting factor in order to highlight the accumulative effects of repeated major disaster events.

Specifically, ARI is constructed from two complementary components:

1) The frequency of major disaster events (defined by our threshold criterion), which captures the recent and recurrent hazard impacts at the latitude-longitude grid scale.

2) The country-level WRI value, which serves as a scaling factor to adjust for differences in exposure and vulnerability across regions. In this way, WRI does not determine the event frequency itself but modulates its relative impact, ensuring that the same number of events in two different countries will not be treated as equivalent if their social and institutional resilience differs.

We acknowledge the referee's concern that the opacity of WRI's internal weighting may introduce interpretive challenges. To address this, we have revised the manuscript to clarify the conceptual role of WRI within ARI: it functions not as a direct additive component, but as a contextual adjustment that reflects underlying socio-economic conditions. Furthermore, ARI is explicitly designed as a complementary measure to WRI rather than a replacement. Whereas WRI provides a holistic but relatively static long-term assessment, ARI emphasizes the short-term accumulative stress induced by multiple disasters, which WRI alone cannot capture.

To improve transparency, we have added a methodological note in the revised manuscript, explicitly stating that: ARI results are primarily driven by event data from EM-DAT, while WRI values serve as modifiers of relative vulnerability. The aggregation to 5°×5° grids is applied only after this adjustment, allowing consistent spatial comparisons. The interpretive focus of ARI should therefore be placed on the patterns of accumulative disaster occurrence, with WRI serving only as a secondary factor contextualizing vulnerability.

We believe this clarification will help distinguish the distinct functions of ARI and WRI and avoid the misunderstanding that ARI is merely a spatial aggregation of WRI values. In future work, we also plan to test the sensitivity of ARI to alternative vulnerability indices, such as INFORM or HDI-based indicators, to further assess the robustness of results under different socio-economic adjustment schemes.

4. Results and interpretation:

4.1. The results are descriptive and primarily spatial distributions of event types. They do not sufficiently engage with the implications of the findings.

We sincerely thank the referee for this constructive comment. We agree that in the original manuscript, the results were largely descriptive, with an emphasis on mapping the spatial distributions of disaster events and ARI values. While these visualizations are informative, we acknowledge that the interpretive depth regarding their broader implications was limited.

To address this concern, we have substantially expanded the new Discussion section of the revised manuscript. Specifically, we now:

• Link ARI patterns to regional socio-economic contexts: We emphasize that the clustering of high ARI grids in Asia—particularly South and Southeast Asia—reflects not only physical hazard exposure but also structural vulnerabilities such as high population density, rapid urbanization, and limited disaster preparedness. This highlights regions where accumulative impacts may overwhelm local coping capacity.

• Highlight policy relevance and decision-making implications: We discuss how ARI can inform priority setting for

international humanitarian aid and disaster risk reduction planning. For instance, regions repeatedly identified as ARI hotspots (e.g., the Himalayan belt and Southeast Asia) may warrant greater investment in resilience-building and recovery assistance, as these areas face persistent accumulative stress rather than isolated events.

 • Situate ARI in relation to existing indices: We point out that traditional long-term indices such as WRI may underestimate risks in areas where recent disaster clustering has significantly eroded resilience. This limitation becomes particularly pronounced within large countries such as China and India, where disaster exposure vary widely across different subnational regions. By comparing ARI with WRI, we reveal cases where discrepancies highlight blind spots in current global risk assessment practices, especially in capturing short-term accumulative impacts at finer spatial scales.

 • Explore scientific implications: We discuss how ARI contributes to ongoing debates about cascading and compounding risks under climate change. The observed geographic clustering suggests that certain regions are entering feedback loops where repeated disasters exacerbate vulnerability, leading to non-linear risk accumulation.

In summary, while our study remains preliminary, we believe the revised manuscript now better engages with the broader implications of the findings. By framing ARI not only as a descriptive spatial tool but also as an instrument to highlight neglected dimensions of accumulated risk, we aim to strengthen its scientific and policy relevance.

*4.2. The conclusions (Line 188) are too strong relative to the evidence provided. The statement that "traditional risk assessment methods may underestimate risks in high-risk areas and overestimate them in low-risk areas" is not convincingly supported by the presented results. A more cautious and better-argued interpretation is required.*

We sincerely thank the referee for pointing out this important issue. We agree that the original formulation of our conclusion—that "*traditional risk assessment methods may underestimate risks in high-risk areas and overestimate them in low-risk areas*"—was expressed too strongly relative to the evidence provided in the manuscript. Our results indeed indicate that the ARI amplifies the contrast between high-risk and low-risk areas compared with the WRI, but we acknowledge that this does not constitute definitive proof of systematic under- or overestimation by traditional indices.

In the revised manuscript, we have therefore adopted a more cautious and better contextualized interpretation. Specifically, we now emphasize that: ARI highlights sharper disparities between regions than WRI, particularly in areas where repeated disaster clustering has significantly eroded resilience. Such discrepancies are especially pronounced in large countries such as China and India, where disaster exposure vary substantially across subnational regions. This suggests that country-level, long-term indices like WRI may sometimes smooth over these localized accumulative impacts of recent disasters, creating blind spots in global risk assessment practices. We explicitly note that these findings are preliminary and require further validation through comparative studies with other risk indices and datasets.

Accordingly, in the revised Conclusions section, we have replaced the previous strong statement with a more cautious and balanced formulation:

"*The comparison between ARI and WRI suggests that long-term, country-level indices may sometimes fail to fully capture the accumulative effects of recent disaster clustering, particularly in subnational regions of large countries where exposure vary widely. This may result in less pronounced contrasts in traditional assessments compared to those revealed by ARI. While this does not imply systematic misestimation by existing methods, it highlights the importance of incorporating accumulative disaster effects and finer spatial grids into future global risk assessment frameworks.*"

We believe this revision better aligns our conclusions with the evidence presented, while still underscoring the conceptual contribution of ARI.

*4.3. A dedicated Discussion section should be added, critically evaluating the robustness of the findings, uncertainties in the data, and implications for disaster risk management.*

We sincerely thank the referee for this constructive suggestion. We fully agree that the manuscript in its original form did not sufficiently provide a critical discussion of the robustness of our findings, the uncertainties associated with the dataset, and

the broader implications for disaster risk management.

In response, we have added a dedicated Discussion section in the revised manuscript. In addition to the previously mentioned content, this new section mainly aims to explicitly address the following aspects:

• Robustness of findings: We discuss how the key spatial patterns identified by the ARI remain consistent even when alternative thresholds (e.g., 40, 60, and 70 fatalities) or different spatial units (e.g., 2.5°×2.5° and 10°×10° latitude-longitude grids) are considered. We also highlight that ARI's comparative role, rather than its absolute values, is the most robust outcome of this exploratory study.

• Uncertainties in the data: We acknowledge the limitations of the EM-DAT database, including missing or incomplete economic loss data, non-standardized georeferencing of disaster locations, and potential reporting biases in earlier years and in developing countries. In addition, we also recognize the uncertainties inherent in the WRI, which integrates multiple dimensions, yet the underlying data quality and weighting scheme are not always fully transparent. Taken together, these uncertainties highlight the need for cautious interpretation of ARI results. They also reinforce our intention that ARI should be viewed not as a definitive measure of disaster risk, but as a complementary tool to existing indices, designed to foreground recent accumulative impacts and to encourage further refinement in global risk assessment practices.

• Implications for disaster risk management: We emphasize that ARI is not intended to replace existing global indices (such as WRI), but rather to complement them by highlighting accumulative disaster impacts at finer geographic scales. This has practical relevance for policymakers and humanitarian organizations, particularly in identifying "hotspot" regions where repeated disasters strain resilience. The integration of ARI outputs with open-source remote sensing datasets (the 3H Dataset) is also discussed as a way to support applied monitoring and early response systems.

By explicitly structuring this new Discussion section, we believe the revised manuscript now provides a more balanced and critical evaluation of both the strengths and limitations of our approach, while clearly outlining its added value and potential applications.

5. Integration of remote sensing data: the role of remote sensing imagery in the analysis is unclear. While the data collection effort is commendable, the manuscript does not explain how these datasets contribute to or validate the ARI results. This section currently feels disconnected from the core analysis.

We sincerely thank the referee for highlighting this important issue. We acknowledge that in the submitted version, the role of remote sensing data may not have been clearly articulated, which could give the impression that this section is disconnected from the ARI analysis.

Our intention in introducing the remote sensing datasets—compiled into the open-source "3H Dataset"—is not to directly validate the ARI at this preliminary stage, but rather to establish a foundation for future methodological integration. Specifically, the ARI is designed to identify spatial clusters of accumulated disaster risks, while the 3H Dataset provides normalized and accessible remote sensing data for those high-risk grids, with a focus on developing countries. By combining these two elements, we aim to create a bridge between macro-level risk assessment and micro-level hazard detection. In practice, the ARI can help to prioritize where remote sensing efforts should be focused (e.g., monitoring urban growth in hazard-prone regions, or tracking recovery in repeatedly affected areas), while the 3H Dataset can support the development and training of remote sensing-based models to detect damage patterns and validate risk assumptions at finer spatial scales.

In the revised manuscript, we have added text to clarify this connection. Specifically, we now explain that the ARI results highlight high-risk areas where accumulative disaster impacts are most pronounced, and the 3H Dataset provides a complementary resource to operationalize and extend these findings through Earth observation techniques. This integration underscores the practical utility of our work: while ARI provides a conceptual and analytical framework for accumulated risk, the 3H Dataset enables future validation and application in disaster monitoring and management, particularly in data-scarce developing regions.

**Minor comments:**

1. Line 107: "*percentages reflect approximate contributions to global damage and losses from 2013 to 2023*" → add a reference.

We appreciate the referee's careful reading and valuable suggestion. We would like to clarify that the percentages reported in Line 107 are not directly derived from a published reference, but are instead calculated by the authors based on the subset of major disaster events identified in EM-DAT for the period 2013-2023. To avoid any ambiguity, we have revised the sentence in the manuscript to explicitly state that these percentages are approximate values derived from our own calculations using EM-DAT data. The revised sentence now reads:

"*A simple calculation based on EM-DAT data reveals that major disaster events corresponding to the four natural hazards account for 9.9%, 22.7%, 58.1%, and 9.3%, respectively. These percentages represent approximate shares of global disaster-related damage and losses during 2013-2023, as estimated from the selected major events.*"

We believe this clarification makes the source and nature of these values more transparent.

2. Line 115: provide a table with examples of how textual descriptions of event locations were translated into latitude/longitude coordinates. This is crucial for validating spatial accuracy, especially when events span multiple grid cells. Clarify how multi-cell events were handled.

We sincerely thank the referee for this helpful suggestion. We agree that illustrating the procedure by which textual descriptions of disaster locations were translated into latitude-longitude coordinates will improve the transparency and reproducibility of our approach. In the revised manuscript, we have added a supplementary table (Table 2) that provides several representative examples. This table shows the original EM-DAT textual description, the selected representative location (e.g., city or administrative unit), and the corresponding latitude-longitude coordinates derived from administrative boundary data.

**Table 2: Example of translating textual descriptions of event locations into latitude-longitude coordinates.**

| Event date | Disaster type | Disaster subtype | Country | Location (textual description & geographic center) | Latitude (°) | Longitude (°) |
|---|---|---|---|---|---|---|
| 2021-8 | Earthquake | Ground movement | Haiti | Nippes Department (18.40° N, 73.00° W) | 18.41* | -73.48* |
| | | | | Grand'Anse Department (18.55° N, 74.00° W) | | |
| | | | | Sud Department (18.35° N, 73.55° W) | | |
| 2019-9 | Storm | Tropical cyclone | Bahamas | Great Abaco Island (26.39° N, 77.09° W) | 26.47 | -77.18 |
| | | | | Grand Bahama Island (26.66° N, 78.32° W) | | |
| 2020-8 | Flood | Flood (General) | Pakistan | Sindh Province (25.50° N, 69.00° E) | 26.35 | 68.85 |
| | | | | Khyber Pakhtunkhwa Province (34.95° N, 72.33° E) | | |
| | | | | Balochistan Province (28.00° N, 67.00° E) | | |
| 2014-8 | Mass movement | Landslide | Nepal | Jure Village (27.77°N, 85.87°E) | 27.77 | 85.84 |
| | | | | Mankha Village (27.76°N, 85.83°E) | | |

*: Precise latitude-longitude coordinates are available for all earthquake events in EM-DAT.

Regarding events that span multiple grid cells, we clarify that our procedure assigns each event to a single representative coordinate. In selecting this coordinate, we prioritize locations that lie near the center of the reported affected area and where the most severe impacts were observed. While this introduces some approximation, it ensures a consistent and standardized allocation across all events, which is essential for global-scale analysis. We explicitly note this limitation in the revised manuscript, and we discuss its implications in the new Discussion section under "Uncertainties in the data".

3. Equation 1: the risk formulation omits the exposure component, which limits its alignment with established risk theory (as WRI dose). This should be acknowledged and discussed.

We thank the referee for this constructive comment. We acknowledge that Equation 1 in the manuscript, expressed as "*Risk*

= *Hazard × Vulnerability*", does not explicitly include the component of exposure. As the referee points out, this omission may appear inconsistent with established formulations of disaster risk, such as the WRI framework.

In fact, as we clarified in the surrounding text, the equation was intended only as a simplified schematic representation to introduce the conceptual basis of disaster risk assessment. We also noted that this basic formulation has evolved in different indices and frameworks, such as the Disaster Risk Index (DRI), which incorporates hazard, population, and vulnerability, and the WRI, which defines risk as the geometric mean of exposure and vulnerability. The purpose of Equation 1 was therefore illustrative rather than prescriptive.

At the same time, we agree with the referee that exposure is indispensable in capturing the realities of disaster risk, particularly when hazard intensity and vulnerability vary spatially but exposure levels determine the magnitude of consequences. To address this concern, we have revised the manuscript to clarify that although Equation 1 does not explicitly contain exposure, the operational framework of the ARI does incorporate it through the use of WRI as a weighting factor. Since the WRI integrates both exposure and vulnerability dimensions, this ensures that ARI captures the combined influence of hazard occurrence and the socio-spatial context of affected regions.

In the revised manuscript, we now explicitly acknowledge in the related section that Equation 1 is a simplified schematic and not a full risk equation, and add a brief discussion noting that exposure is indirectly embedded in the ARI framework via the WRI component, while also recognizing that this indirect representation may limit interpretability compared to approaches where exposure is modeled explicitly.

4. Chapter 3 (supplement of remote sensing data): clarify its role in the study. does it serve as validation, contextual information, or a complementary dataset? Currently, its integration is not explained.

We thank the referee for raising this important point. We agree that the role of the "supplement of remote sensing data" in Chapter 3 was not sufficiently explained in the original version.

To clarify, the remote sensing component of our study consists of two complementary parts, as reflected in the chapter title "*Integration and supplement of remote sensing data*":

• Integration of remote sensing data: We conducted an extensive review of existing open-source datasets, which provide valuable sub-meter satellite images, but their coverage is uneven. They tend to prioritize major cities or certain regions, while many high-risk grids identified by ARI—particularly in developing countries—remain underrepresented. Moreover, available datasets differ in resolution, image type, and preprocessing standards, which complicates their direct application.

• Supplement of remote sensing data: To address these gaps, we supplemented high-resolution visible spectral imagery specifically for two high-risk grids, where no relevant open-source data was available. For each of these grids, we acquired and standardized sub-meter imagery covering a 50 km² area (100 km² in total), centered on the coordinates of the disaster event with the maximum "Total Deaths". The supplemented imagery was standardized and then integrated with the existing open-source datasets. This supplementation ensures that the most data-scarce yet high-risk grids are adequately represented.

Importantly, this remote sensing dataset—compiled as the 3H Dataset—is not intended to directly validate the ARI in this preliminary stage. Rather, it complements the ARI framework by providing a foundation for future methodological integration. Specifically, the ARI identifies spatial clusters of accumulated disaster risk, while the 3H Dataset provides accessible remote sensing resources for those high-risk grids. Together, they establish a bridge between macro-level risk assessment and micro-level hazard detection, enabling future applications such as validating ARI patterns with Earth observation data, monitoring post-disaster recovery, and training machine learning models for damage detection.

In the revised manuscript, we have expanded Chapter 3 to emphasize this complementary role and to clarify how integration and supplement jointly contribute to building a usable dataset that operationalizes ARI findings.

Once again, we sincerely appreciate your positive comments and valuable suggestions on our manuscript. We are grateful for the opportunity to improve the manuscript and hope the revised manuscript has now met the publication standard of your journal.

Kind Regards,

Qingzhao Kong and Erqi Zhu
25th Aug. 2025